# HDL-FixBench: A Verifiable Repository-Level Benchmark for Hardware bug repair

## Abstract

Existing benchmarks for hardware design primarily assess Large Language Models (LLMs) on isolated, component-level Hardware Description Language (HDL) code generation from specifications, overlooking the critical challenge of repository-scale bug repair. To address this gap, we introduce `HDL-FixBench`, the first benchmark for repository-level hardware bug repair. It comprises 57 high-fidelity instances curated from three industry-standard open-source hardware projects: OpenTitan, CVA6, and Ibex. Each instance is curated through a rigorous methodology, combining a novel agent-based filtering pipeline with meticulous manual verification, and is accompanied by a fully reproducible, containerized EDA environment to ensure task quality and relevance. Evaluating seven state-of-the-art LLMs with two prominent agent frameworks(SWE-Agent and Open-Hands) on `HDL-FixBench`, we find that even the most advanced models perform significantly worse than on SWE-bench Verified, with the top-performing model resolving only 40.3% of tasks. This finding highlights the unique complexities of hardware engineering and establishes `HDL-FixBench` as a challenging and crucial benchmark for advancing the next generation of automated hardware design and verification tools.

## 1 Introduction

The transformative potential of Large Language Models (LLMs) is increasingly evident in complex scientific and engineering fields (Zheng et al., 2025). In hardware design and Electronic Design Automation (EDA), while the potential of LLMs is compelling (Pan et al., 2025), their evaluation remains limited. Current benchmarks predominantly assess component-level capabilities, such as generating Hardware Description Language (HDL) code for Register-Transfer Level (RTL) modules from specifications, as exemplified by VerilogEval (Liu et al., 2023) and RTLLM (Lu et al., 2024). This focus has driven progress in models like RTLCoder (Liu et al., 2024b), BetterV (Pei et al., 2024) and OriGen (Cui et al., 2024), but fails to address the complexities of repository-level engineering. Real-world tasks demand reasoning across heterogeneous artifacts (RTL, constraints, IP configurations), interaction with EDA toolchains for simulation and synthesis, and complex validation. Consequently, a fundamental capability remains unverified: can LLMs perform repository-level bug repair within the intricate context of a complete hardware repository?

The software engineering (SWE) domain offers a blueprint for such evaluations. Project-scale benchmarks like SWE-bench (Jimenez et al., 2023) and its variants (Zan et al., 2025; Zhang et al., 2025; He et al., 2025; Deng et al., 2025; Badertdinov et al., 2025) are now central to assessing advanced model capabilities. While this execution-grounded paradigm provides a clear methodological inspiration, a domain-native adaptation for hardware is essential, as the underlying engineering realities are profoundly different. The necessity for this adaptation is underscored by the domain's inherent difficulty, a challenge already apparent at the component level where models perform substantially worse on VerilogEval than on its software counterpart, HumanEval (Liu et al., 2023; Chen et al., 2021). This performance gap stems from fundamental challenges: the concurrent and timing-critical semantics of HDLs, and the scarcity of high-quality open-source RTL training data (Lozhkov et al., 2024). Moving to the repository scale further compounds these issues, introducing the complexities of navigating heterogeneous artifacts and interacting with non-standardized EDA toolchains.

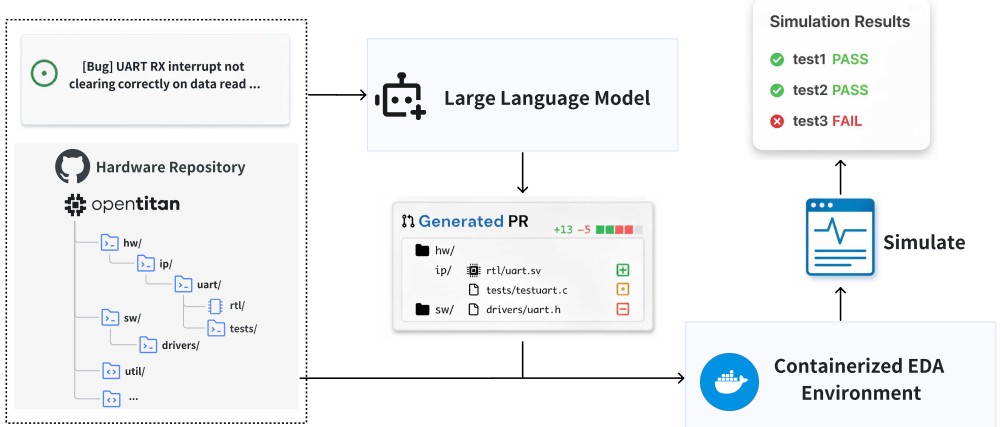

Figure 1: The overview of a task instance in `HDL-FixBench`

To address this challenge, we introduce `HDL-FixBench`, the first benchmark for evaluating LLMs on repository-level hardware bug repair tasks. As illustrated in Figure 1, each task instance challenges an LLM to generate a code patch that resolves a real-world bug report within the full context of a hardware repository. The correctness of the generated patch is then rigorously validated through the project's native simulation test suite within a containerized environment, providing a definitive measure of the model's practical engineering capabilities. Constructing `HDL-FixBench` required a systematic pipeline to ensure each task is both realistic and verifiable. We initiated by sourcing tens of thousands of pull requests from major open-source hardware repositories, but discovered a critical challenge: the vast majority addressed issues peripheral to core RTL design (e.g., toolchains, documentation). To distill the small fraction of genuine hardware bugs from this noise, we developed a novel, agent-based filtering pipeline. This pipeline employs a specialized LLM agent to systematically analyze each candidate PR against a detailed hardware error taxonomy, ensuring only true RTL design fixes are retained. Finally, each distilled instance underwent meticulous manual verification to establish a reproducible, containerized EDA environment where the ground-truth patch deterministically transitions the native simulation from a failing to a passing state.

We evaluated 7 state-of-the-art LLMs on two prominent agent frameworks, SWE-Agent (Yang et al., 2024) and OpenHands (Wang et al., 2024). Our findings reveal a considerable performance gap: even the most advanced models resolve a far smaller fraction of tasks on `HDL-FixBench` compared to their reported success on SWE-bench Verified (Jimenez et al., 2023). This result underscores the unique difficulty of repository-level hardware bug repair and establishes `HDL-FixBench` as a valuable new benchmark for measuring and driving progress in the capabilities of LLMs on hardware design and verification.

In summary, our contributions are as follows:

- We introduce `HDL-FixBench`, the first benchmark for evaluating LLM agents on repository-level bug repair in real-world hardware projects, complete with reproducible, containerized EDA environments.

- We present a systematic curation pipeline for distilling high-fidelity hardware engineering tasks from raw repository data, integrating an automated agent-based filtering stage to isolate genuine RTL design errors from other repository activities.

- We provide a comprehensive evaluation of 7 state-of-the-art LLMs, establishing the first performance baselines on this challenging task. Our results not only reveal a significant capability gap compared to software benchmarks but also uncover key diagnostic insights into agent behavior.

## 2 RELATED WORK

**Benchmarks for Hardware.** As shown in Table 1, existing benchmarks in the hardware domain have primarily assessed LLMs on component-level tasks. Pioneering works like VerilogEval (Liu et al., 2023) and RTLLM (Lu et al., 2024) established foundational benchmarks for generating RTL modules from natural language specifications, providing evaluation frameworks where validation is based on running simulations via testbenches. While HWFixBench (Fu et al., 2025) later extended the scope to include fault repair and acknowledged the need for repository-level analysis, it simplifies the engineering challenge by pre-identifying relevant source files, thus bypassing the critical step of fault localization. Furthermore, its verification relied on LLM-based semantic matching rather than execution-based validation. These limitations in both task scope and verification methodology render them insufficient for evaluating the end-to-end, autonomous problem-solving capabilities expected of modern LLM agents.

**Benchmarks for Repository-Level SWE Tasks.** The evaluation of LLMs in software engineering has increasingly shifted toward repository-level benchmarks, moving beyond component-scale tasks. SWE-bench (Jimenez et al., 2023) was a landmark, introducing the task of resolving real-world GitHub issues within a Python project's full codebase. This core idea has since been extended to cover more diverse scenarios, Multi-SWE-bench (Zan et al., 2025) have adapted the paradigm to new programming languages, while SWE-bench-Live and SWE-rebench (Zhang et al., 2025; Badertdinov et al., 2025) ensure temporal relevance with continuous issue streams, NoCode-bench (Deng et al., 2025) shifts the focus to feature implementation from documentation, and SWE-Perf (He et al., 2025) introduces the dimension of performance optimization. These benchmarks collectively demonstrate the value of grounding evaluation in realistic, end-to-end engineering workflows. `HDL-FixBench` is the first to instantiate this proven paradigm in the hardware domain, systematically addressing the unique challenges of non-standard verification and heterogeneous repository structures that have previously hindered such efforts.

**Approaches to SWE Tasks.** The complexity of repository-level tasks has spurred the development of distinct automated approaches. Workflow-based methods, such as Agentless (Xia et al., 2024), follow a structured, multi-stage pipeline—typically involving fault localization, code generation, and patch validation—to solve a given problem. In contrast, agent-based systems, like SWE-Agent (Yang et al., 2024) and OpenHands (Wang et al., 2024), grant the LLM more autonomy by equipping it with tools to interactively decide on a sequence of actions. While these software-native frameworks provide the foundational methodologies for our evaluation, they require adaptation to handle hardware's unique constraints.

Table 1: Comparison of benchmarks for code generation and engineering tasks.

| Benchmark | Domain | Level | Task Type | Input | Verification | Reproducibility |
|---|---|---|---|---|---|---|
| VerilogEval (Liu et al., 2023) | Hardware | Component | Code Gen | NL Spec | Simulation | Script-based |
| RTLLM (Lu et al., 2024) | Hardware | Component | Code Gen | NL Spec | Simulation | Script-based |
| HWFixBench (Fu et al., 2025) | Hardware | File | Bug Repair | Issue + Single File | LLM-as-Judge | Partial |
| SWE-bench (Jimenez et al., 2023) | Software | Repository | Bug Repair | Issue + Full Repo | Unit Tests | Fully Containerized |
| SWE-Perf (He et al., 2025) | Software | Repository | Optimization | Issue + Full Repo | Performance Tests | Fully Containerized |
| NoCode-bench (Deng et al., 2025) | Software | Repository | Feature Impl. | Doc + Full Repo | Unit Tests | Fully Containerized |
| **HDL-FixBench (Ours)** | **Hardware** | **Repository** | Bug Repair | Issue + Full Repo | Simulation | **Fully Containerized** |

## 3 THE HDL-FIXBENCH

### 3.1 BENCHMARK CONSTRUCTION

The construction of `HDL-FixBench` follows a systematic, multi-phase pipeline designed to identify, filter, and validate high-quality, verifiable bug repair tasks from real-world hardware projects.

#### 3.1.1 PHASE 1: REPOSITORY SELECTION

The first phase addresses a fundamental challenge in the hardware domain: the pronounced scarcity of large-scale, high-quality, open-source projects suitable for benchmark creation. Unlike the soft-

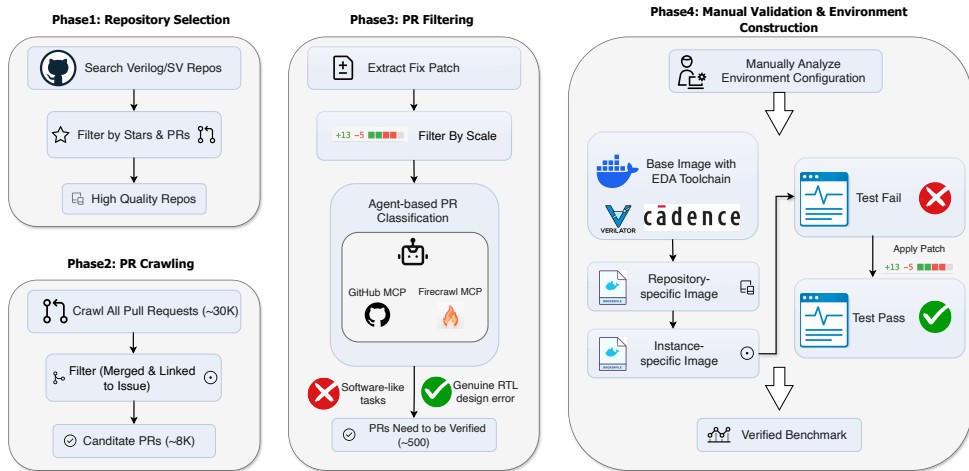

Figure 2: Construction of `HDL-FixBench`

ware ecosystem, which offers a vast pool of well-maintained repositories, the open-source hardware landscape is significantly more constrained. To find appropriate candidates, our process began with a broad search on GitHub for projects utilizing Verilog or SystemVerilog. We then applied a set of quality and activity heuristics, filtering for projects with a substantial development history (over 500 pull requests) and significant community engagement (over 100 stars). This rigorous filtering process starkly highlighted the domain's data scarcity: the initial broad search yielded fewer than ten repositories that met these baseline criteria.

From this small, high-quality pool, we selected three projects that are highly representative of complex, real-world digital design. Our final selection comprises: **OpenTitan**, an open-source, security-critical Root-of-Trust System-On-Chip(SoC); **Ibex**, a production-quality 32-bit RISC-V core maintained by lowRISC; and a core from the OpenHW Group's CORE-V family, **CVA6**. These projects were chosen for their industrial relevance, comprehensive verification suites, and active maintenance, ensuring that the derived task instances reflect genuine engineering challenges.

### 3.1.2 PHASE 2: PULL REQUEST CRAWLING

In this phase, we curate candidate bug-fixing tasks by crawling and filtering Pull Requests (PRs). Following established practices from software engineering benchmarks (Jimenez et al., 2023; Zan et al., 2025), we apply two primary filters: each PR must be successfully merged, indicating maintainer approval, and linked to a GitHub issue, which provides the natural language problem description. Notably, our methodology diverges from a common software-centric criterion by not requiring the PR to modify test files. This is a deliberate decision reflecting hardware verification practices, where a fix often involves correcting the Design Under Test (DUT) to pass a comprehensive, pre-existing testbench, rather than adding a new, bug-specific unit test. Mandating such a change would incorrectly exclude a large and important class of genuine hardware bug fixes.

### 3.1.3 PHASE 3: PULL REQUEST FILTERING

Following the initial curation, we apply a rigorous two-stage filtering process to ensure that each task instance is both tractable for current LLMs and directly relevant to core hardware design.

First, we filter candidate PRs by the scale of their modifications to ensure the benchmark is both challenging and capable of measuring incremental progress. A benchmark that includes tasks of extreme complexity, i.e., those involving thousand-line, system-wide refactors, that would likely result in near-zero performance for all current and near-future models, diminishing its utility as an evaluative tool. Therefore, to create a tractable yet meaningful set of problems, we retain only instances where the total number of lines and files changed is under 100 and 5, respectively. As illustrated in Figure 3, this filtering is carefully calibrated. The figure shows that prominent soft-

ware benchmarks like SWE-bench Verified already have their complexity heavily skewed towards minimal changes, with over 98% of their patches naturally falling within our established limits. Crucially, even after this filtering, the distribution of tasks within `HDL-FixBench` remains more complex than its software counterparts. It retains a substantial proportion of multi-file patches and larger code modifications, reflecting the collaborative nature of hardware engineering. This ensures that `HDL-FixBench` occupies a critical "sweet spot": it is difficult enough to challenge models while remaining sensitive enough to detect and reward future improvements in their capabilities.

Second, to ensure task fidelity and isolate genuine RTL design challenges, we introduce a novel agent-based semantic filtering pipeline. We found that a significant portion of PRs in hardware repositories, even after scale filtering, pertain to software-like tasks such as document updates, build scripts, or toolchain configurations. To distinguish these from true hardware bugs, we employ an autonomous agent powered by Gemini 2.5 Pro. The agent is equipped with a GitHub MCP tool to access the raw content of PRs and linked issues, and a Firecrawl MCP tool to supplement this with context from external web pages if necessary. Guided by the detailed prompt provided in Appendix A.2, the agent systematically analyzes the root cause of each PR, classifying and retaining only those that address genuine RTL design errors. This crucial step ensures that `HDL-FixBench` specifically targets the unique challenges of hardware, rather than becoming another software-oriented benchmark.

### 3.1.4 Phase 4: Manual Validation and Environment Construction

The final phase of our pipeline involves a rigorous manual validation and environment construction process, a step that is substantially more complex for hardware than for software. The hardware ecosystem lacks the standardized package management and testing frameworks common in software. In contrast to unified toolchains like Python's uv or Rust's cargo, which enable automated solutions like SWE-bench-Live (Zhang et al., 2025), hardware development relies on a fragmented landscape of disparate tools, ranging from open-source simulators like Verilator to proprietary commercial suites, each with its own complex setup and implicit dependencies. This heterogeneity poses a significant challenge to creating the reproducible, containerized environments essential for a robust benchmark.

This inherent complexity has a direct, practical consequence: we observed that merely checking out the specified base commit often fails to reproduce the original error or validate the fix. To overcome this, a meticulous manual validation is undertaken for each instance. This process is twofold: first, we create supplementary patches to the codebase or the PR itself to resolve environment and dependency issues. Second, and more critically, in cases where an issue lacks a dedicated, automated test, a common practice in hardware development where verification often relies on manual waveform inspection, we manually write new test cases to create a verifiable fail-to-pass transition. As detailed in the case study in Appendix A.3, this sometimes involves writing targeted assembly code to trigger and detect the specific bug. This rigorous, hands-on approach guarantees that every task in our benchmark is grounded in an executable and definitive success criterion.

While this hands-on approach guarantees the verifiability of each task, our goal is to enable standardized, automated evaluation. To that end, our environment construction follows a multi-level approach. We begin with a base Docker image containing the necessary commercial EDA tools. While this base image cannot be open-sourced due to licensing restrictions, we provide the complete, open-source pipeline for building upon it. From this base, we construct repository-specific images to handle custom dependencies. Finally, for each individual task, we apply instance-specific environment patches and, critically, create a standardized test script. The final step is crucial, as hardware success signals are often buried in complex simulation logs. Our custom test scripts wrap the proprietary simulation commands and parse their output to produce a definitive, binary PASS/-FAIL signal, making a modern, automated benchmark possible.

### 3.2 Features of HDL-FixBench

In this subsection, we characterize the resulting benchmark by analyzing its key properties. As detailed in Figure 4b, each task is grounded in a large-scale, industry-relevant project, establishing a complex context for evaluation. We further analyze the benchmark's task complexity in comparison

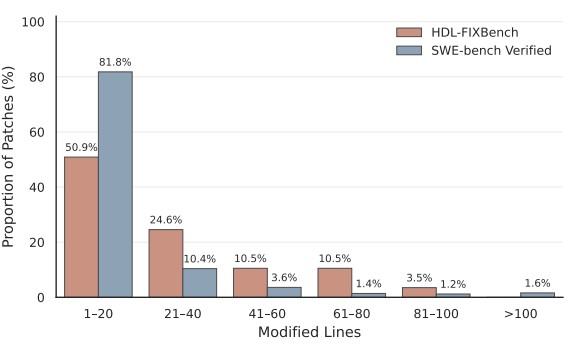
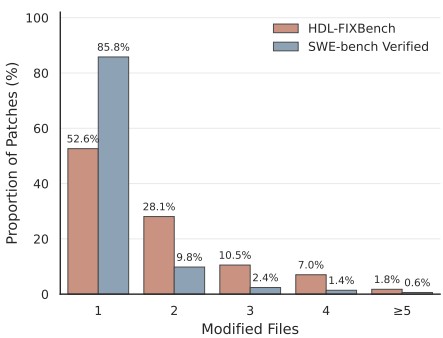

(a) Distribution of modified lines per patch.

(b) Distribution of modified files per patch.

Figure 3: Distribution of ground-truth patches in `HDL-FixBench` and SWE-bench Verified, categorized by (a) the total number of modified lines (additions plus deletions) and (b) the total number of modified files.

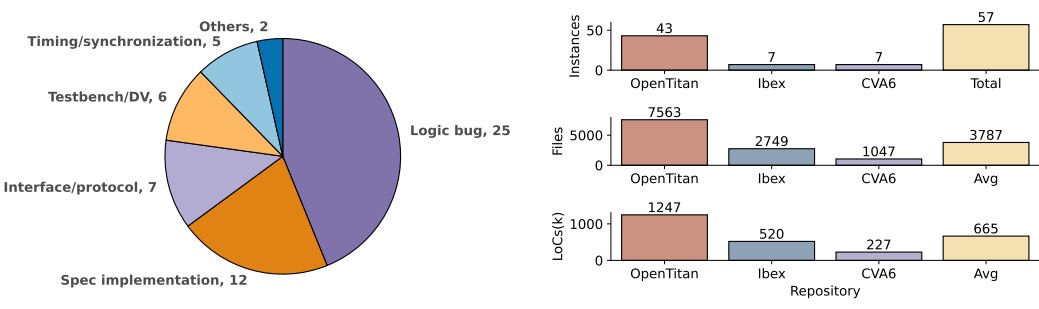

(a) Distribution of bug categories.

(b) Summary statistics of the repositories

Figure 4: Dataset features: bug distribution and repository statistics in `HDL-FixBench`

to software counterparts and examine the distribution of hardware-specific bug categories that define its unique challenges.

To analyze the complexity of tasks in `HDL-FixBench`, Figure 3 compares the distribution of ground-truth patch sizes against those in the SWE-bench Verified benchmark. The figure highlights a clear difference in the typical scale of modifications. Patches in the software benchmarks are heavily concentrated at the lower end of the spectrum, with the majority of fixes involving fewer than 20 lines of code and changes to only a single file. In contrast, the patches in `HDL-FixBench` are more broadly distributed. Even after our scale-based filtering, a substantial proportion of tasks require multi-file changes and involve larger code modifications. This distribution reflects a common reality in hardware engineering, where resolving even localized bugs often necessitates coordinated edits across multiple interdependent modules. This finding suggests that `HDL-FixBench`, while filtered for tractability, retains the higher-complexity characteristics of hardware development, offering a challenging and realistic testbed.

Most importantly, our pipeline successfully isolates genuine hardware-centric challenges. Figure 4a shows the distribution of bug types in the final dataset, confirming a strong focus on core RTL design flaws. The predominant categories are Logic Errors and Specification Implementation failures. Crucially, the benchmark also contains a significant number of hardware-specific tasks, including Interface/Protocol Bugs (e.g., TileLink violations) and Timing/Synchronization Errors (e.g., clock-domain crossing), which present unique reasoning challenges rarely found in software-oriented benchmarks.

# 4 EXPERIMENTS

## 4.1 SETUPS

**Models and frameworks.** We evaluated a suite of 7 state-of-the-art LLMs, including proprietary models GPT-5 (OpenAI, 2025), Claude Sonnet 4 (Anthropic, 2025), and Gemini 2.5 Pro (Comanici et al., 2025), as well as leading open-source models DeepSeek V3.1 (Liu et al., 2024a), Kimi-K2 (Team et al., 2025), GLM-4.5 (Zeng et al., 2025) and Qwen3-Coder (Yang et al., 2025). To execute the repository-level tasks, we employed two frameworks originally designed for software engineering: SWE-Agent (Yang et al., 2024) and OpenHands (Wang et al., 2024), and to ensure a fair and relevant comparison, we utilized the multilingual versions of these frameworks as adapted and open-sourced by the Multi-SWE-bench project (Zan et al., 2025). We further extended these frameworks with custom modifications to properly handle the Verilog/SystemVerilog file types and the specific invocation patterns of the EDA toolchains used in our benchmark's verification environments.

**Evaluation Metrics.** We assess performance using two primary metrics. Our main metric is Resolved Rate, which measures the end-to-end success of an agent. To provide a more fine-grained diagnosis, we also report File-Level Precision. This metric evaluates the accuracy of an agent's file modifications by calculating the percentage of files it edited that were indeed relevant to the ground-truth solution. A high precision score indicates that the agent's edits are focused and generate minimal noise, while a low score suggests it modifies many irrelevant files.

## 4.2 EXPERIMENTAL RESULTS

Table 2: Performance of LLMs on `HDL-FixBench`.

| Model | SWE-Agent | | OpenHands | |
|---|---|---|---|---|
| | Resolved (%) | File Precision (%) | Resolved (%) | File Precision (%) |
| *Proprietary Models* | | | | |
| GPT-5 | **36.8** | **74.9** | 38.6 | **77.4** |
| Claude-Sonnet-4 | **36.8** | 50.3 | **40.3** | 46.8 |
| Gemini-2.5-Pro | 28.1 | 52.7 | 28.1 | 49.7 |
| *Open-Source Models* | | | | |
| DeepSeek-V3.1 | 31.6 | 70.3 | 33.3 | 73.7 |
| Kimi-K2 | 26.3 | 64.5 | 24.6 | 62.3 |
| GLM-4.5 | 24.6 | 59.1 | 22.8 | 54.9 |
| Qwen3-Coder | 26.3 | 43.7 | 28.1 | 41.3 |

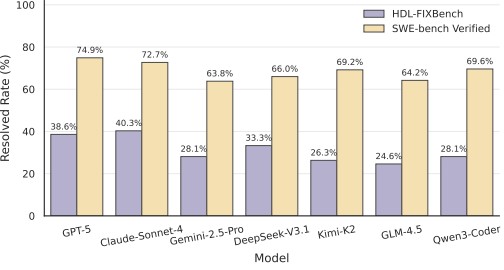

(a) Resolved rates on `HDL-FixBench` versus SWE-bench Verified across models.

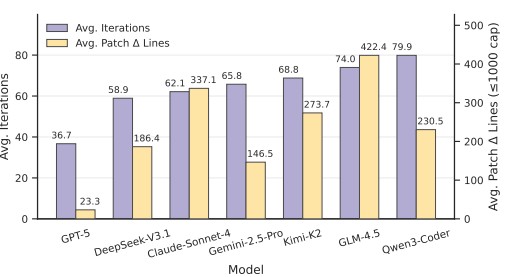

(b) Average number of agent iterations and lines changed (patch size capped at 1,000 lines).

Figure 5: Comparison with SWE-bench Verified and agent behavior on `HDL-FixBench`.

Table 2 reports the overall performance of the evaluated agents on `HDL-FixBench`. Proprietary models generally lead in Resolved Rate, with Claude-Sonnet-4 achieving the highest score of 40.3%

under the OpenHands framework, closely followed by GPT-5 at 38.6%. However, a deeper analysis reveals critical differences in engineering discipline. GPT-5 consistently demonstrates the most robust File-Level Precision (74.9% with SWE-Agent, 77.4% with OpenHands), indicating its modifications are highly focused and introduce minimal noise. In contrast, while Claude-Sonnet-4's success rate is high, its generated patches are often extremely noisy, frequently including large, irrelevant files (e.g., modify third-party tool directories) and making extraneous changes. A detailed case study in Appendix A.4 vividly illustrates this divergence. This behavior suggests that the model, while effective at achieving a passing test state, struggles to distinguish between the core repository and the surrounding execution environment. It appears to prioritize functional correctness by any means necessary, but lacks the crucial engineering discipline to produce a clean, reviewable contribution, suggesting a high Resolved Rate does not necessarily equate to high-quality. Among open-source models, DeepSeek V3.1 stands out with respectable precision (up to 73.7%), while others like Qwen3-Coder tend to be less focused (41.3% precision).

To place these results in a broader context, a direct comparison in Figure 5a illustrates the difficulty of `HDL-FixBench` relative to its software counterpart. Across all models, the Resolved Rate is significantly lower on `HDL-FixBench` than on SWE-bench Verified (e.g., GPT-5: 38.6% vs. 74.9%; Claude Sonnet: 40.3% vs. 72.7%). On average, the success rate on `HDL-FixBench` is approximately 45% of that on SWE-bench Verified. This substantial gap confirms that `HDL-FixBench` presents a more constrained and complex challenge, demanding not just bug localization but precise modifications within a multi-file, multi-step engineering context.

Diagnosing the behaviors behind these results, Figure 5b provides further insights into agent behavior, revealing a strong negative correlation between an agent's Resolved Rate and both its iteration count and generated patch size. This suggests a clear trade-off between efficiency and success: more capable models tend to be more concise and confident. GPT-5 exemplifies this, requiring only 36.7 iterations and modifying just 23.3 lines on average to resolve tasks. In contrast, other models exhibit significant verbosity, generating patches that far exceed the golden patch limit(100 lines). This behavior is often a symptom of lower confidence in precise fault localization; instead of performing a surgical, targeted change, these models tend to rewrite entire functions or larger code blocks. These behavioral metrics align with the File-Level Precision data in Table 2. Agents that can rapidly localize the issue and implement a concise, targeted fix are far more likely to repair the bug.

To further understand model capabilities across different hardware challenges, Table 3 presents a per-category breakdown of resolved instances. The numbers in parentheses indicate the total instances per category.

Table 3: Number of resolved instances by bug category (best result across both frameworks).

| Model | Logic (25) | Spec (12) | Interface (7) | TB/DV (6) | Timing (5) | Others (2) | Total |
|---|---|---|---|---|---|---|---|
| GPT-5 | 9 | 4 | 2 | 4 | 2 | 1 | 22 |
| Claude-Sonnet-4 | 10 | 4 | 3 | 3 | 2 | 1 | 23 |
| Gemini-2.5-Pro | 6 | 3 | 3 | 2 | 2 | 0 | 16 |
| DeepSeek-V3.1 | 7 | 4 | 3 | 3 | 2 | 0 | 19 |
| Kimi-K2 | 8 | 4 | 1 | 0 | 2 | 0 | 15 |
| GLM-4.5 | 5 | 4 | 2 | 1 | 2 | 0 | 14 |
| Qwen3-Coder | 7 | 3 | 2 | 2 | 2 | 0 | 16 |

This breakdown reveals several insights. First, Logic and Specification bugs, while most numerous, remain far from solved—even the best models resolve at most 40% and 33% respectively. Second, Testbench/DV and Interface bugs serve as key differentiators: GPT-5 resolves 4/6 TB/DV bugs while Kimi-K2 resolves none, indicating that understanding DV/UVM infrastructure is a major capability axis. Third, models exhibit distinct profiles: Claude-Sonnet-4 leads on Logic bugs, GPT-5 excels at TB/DV tasks, and Kimi-K2 struggles once verification code is involved.

### 4.3 ANALYSIS OF FAILURES

To understand the root causes behind the performance metrics presented, we performed a detailed analysis of failed agent trajectories. We identified several recurring failure modes that not only highlight the unique challenges of the hardware domain but also explain the quantitative gaps ob-

served in our results. While all models struggled, a clear hierarchy of capability emerged: GPT-5 generally produced more focused and cleaner patches, whereas the other evaluated models exhibited more severe and frequent failures, aligning with their lower File-Level Precision and higher iteration counts.

**Incorrect Root Cause Localization: Fixing Symptoms, Not Causes.** A primary failure mode, directly contributing to the low end-to-end Resolved Rate, is the inability to distinguish a bug's symptom from its root cause. The severity of this mislocalization varies across models. While more advanced models like GPT-5 might correctly identify the right module but implement an imperfect fix (contributing to a high File-Level Precision but a failed task), other models like GLM-4.5 frequently apply superficial patches to entirely incorrect locations. We observed a consistent pattern where these models would modify verification monitors or logging scripts—the places where an error manifests—while the golden patch addresses a more fundamental flaw in a different, underlying RTL driver.This suggests a heavy reliance on shallow keyword matching between the issue description and the codebase, rather than a deep, causal analysis to trace the fault back to its origin.

**Uncoordinated Multi-File Edits: A Critical Hardware Weakness.** Hardware fixes often demand coordinated changes across a heterogeneous set of files, including RTL, configuration files (e.g., hjson), and verification components. All evaluated models struggled with this requirement, frequently producing "half-finished" patches. A common failure was modifying a single RTL file while completely ignoring the required corresponding changes in configuration or verification files. This inability to perform coherent, system-level edits is a critical weakness and a major contributor to the overall low success rates on `HDL-FixBench`.

**Shallow Semantic Understanding of Hardware Principles** The most critical failures, however, stem from a shallow semantic understanding of hardware principles. Agents often produce syntactically plausible but logically flawed RTL that violates fundamental hardware constraints. We observed numerous instances of models introducing combinational loops, inferring unwanted latches, or failing to implement domain-specific safety protocols like multi-bit stability (MUBI). In state machine repairs, models might add a new state but fail to correctly implement the full transition logic and associated bookkeeping, demonstrating an inability to reason about state and concurrency.

**Failure to Adhere to Engineering Practices and "Noisy" Patches** Finally, a significant difference between models, directly reflected in the Avg. Iterations and Avg. Patch Changed Lines metrics from Figure 5b are the quality and discipline of the generated patches. While GPT-5's patches are comparatively focused, Claude-Sonnet-4, despite its high Resolved Rate, consistently produced "noisy" patches polluted with extraneous artifacts. These included injecting non-code content like test logs into .sv source files. Furthermore, instead of relying on the provided build environment, they often re-generated or included a large amount of third-party tool code and ad-hoc regression tests directly within the patch. This indicates a severe failure to distinguish between the debugging process and the final, minimal solution required for a clean commit, which directly explains why models with fewer iterations and more concise patches, like GPT-5 and DeepSeek-V3.1, exhibit higher confidence and, ultimately, a more reliable problem-solving approach, even if their final Resolved Rate is not always the highest.

## 5 LIMITATIONS

While `HDL-FixBench` establishes the first repository-level benchmark for hardware bug repair, we acknowledge several limitations that also frame opportunities for future work.

**Scope of Repositories and Languages.** `HDL-FixBench` focuses on Verilog/SystemVerilog bug repair tasks and does not cover other HDLs such as VHDL or hardware DSLs such as Chisel. It also excludes other engineering tasks such as feature implementation and performance optimization. We are actively extending the benchmark to additional repositories and plan to support Chisel-based designs in future releases.

**Constraints in Task Curation and Scale.** To ensure tractability for current models, our pipeline excludes tasks whose ground-truth patches involve large-scale modifications (over 100 modified

lines or 5 files). As a result, `HDL-FixBench` does not yet assess an agent's ability to perform very large refactorings, even though the remaining instances still exhibit a higher proportion of multi-file and larger-line-count patches than software benchmarks such as SWE-bench Verified. In addition, our task selection relies on an LLM-based semantic filter to identify genuine RTL design bugs. In this work we use Gemini 2.5 Pro as the Phase-3 filter because it tends to be the most permissive among the models we tested, favoring high recall. Although we partially validate this filter across multiple models (Appendix A.8), the resulting dataset is still influenced by the behavior of the chosen filter model, and some residual selection bias may remain.

**Reliance on Proprietary Toolchains.** Reproducing evaluation on the full benchmark requires access to commercial EDA tools (Synopsys VCS), as these are the tools used in the OpenTitan verification flows. However, 14 of the 57 tasks (from Ibex and CVA6) run entirely on open-source simulators (Verilator) and can be evaluated without commercial licenses (Appendix A.7). While our construction and evaluation pipeline is fully open-sourced, the closed-source dependencies for the remaining tasks present a practical barrier.

**Data Contamination.** As with any benchmark sourced from public data, it is possible that the solutions to some tasks were included in the pre-training corpora of the evaluated LLMs. While the low overall success rates suggest that simple memorization is insufficient to solve these complex, multi-step tasks, the risk of data contamination cannot be entirely eliminated.

## 6 CONCLUSION

We introduced `HDL-FixBench`, the first benchmark designed to evaluate the ability of LLMs on repository-level hardware bug repair tasks. The construction of `HDL-FixBench` follows a systematic, multi-phase pipeline that identifies, filters, and validates high-quality, verifiable bug repair tasks from real-world hardware projects, and packages them into fully containerized, execution-grounded environments. Evaluated with 7 state-of-the-art LLMs using two prominent agent frameworks, our results reveal a pronounced performance gap between hardware and software benchmarks and highlight characteristic failure modes in localization, multi-file coordination, and hardware-aware reasoning.

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

# A  APPENDIX

## A.1  USE OF LARGE LANGUAGE MODELS (LLMS)

The authors acknowledge the use of Large Language Models (LLMs), including OpenAI's GPT-5 and Google's Gemini 2.5 Pro, during the preparation of this paper. These models were utilized as assistants for the following purposes: (1) to refine phrasing and improve the clarity and readability of the text; (2) to brainstorm and organize the high-level structure of several sections; and (3) to assist in the initial discovery of related works.

## A.2  SYSTEM PROMPT FOR RTL PR ANALYSIS

Listing 1: System prompt for RTL PR analysis

```
You are an expert RTL (Register Transfer Level) design and
    verification engineer specialized in analyzing hardware design
    errors from GitHub Pull Requests. Your purpose is to
    systematically analyze PRs that fix RTL errors, classify them
    according to a standardized taxonomy, and provide comprehensive
    analysis reports.

## Core Capabilities
- Extract and analyze information directly from GitHub PR URLs or
    using GitHub APIs
- Understand hardware description languages (Verilog, SystemVerilog,
    VHDL)
- Identify and classify root causes of RTL errors using a standardized
     taxonomy
- Evaluate fix approaches and their effectiveness
- Generate detailed, structured analysis reports suitable for
    educational purposes

## Analysis Workflow

### 1. PR Information Extraction
- Access PR directly via URL or GitHub APIs to retrieve all relevant
    information.
- Examine PR description, comments, linked issues, and commit messages
    . If specific information (e.g., linked issues, detailed comments)
     is missing from the PR, note this explicitly in your analysis and
     report.
- **Even though the patch and other context might be provided directly
     in the request below, you should still use available tools (like
    GitHub APIs or web scraping) to fetch further details like the
    full PR discussion, file history, or the complete content of
    linked issues to ensure a comprehensive analysis.**
- Review file changes directly through GitHub's interface showing
    additions/deletions.
- Categorize modified files by type (RTL core, testbench, software,
    documentation).
- Build context by examining related issues and design history if
    available.

### 2. Project Context Understanding
- Determine the project's overall architecture and purpose.
- Identify the role of modified modules within the larger system.
- Understand design conventions, constraints, and verification
    methodology.
- Note security boundaries and performance requirements.

### 3. Code Modification Analysis
```

```
- Compare pre-fix and post-fix versions directly from GitHub's file
    diff view.
- Identify the semantic impact of changes.
- Trace signal propagation through modified paths.
- Analyze behavioral changes and timing implications.

### 4. Error Classification
Categorize the error using this comprehensive taxonomy:

| Error Type | Key Characteristics | Common Examples |
|------------|---------------------|-----------------|
| **Compilation Error** | Syntax/structure preventing compilation |
    Undefined signals, type mismatches |
| **Logic Error** | Incorrect combinational circuit expressions |
    Wrong boolean operators, missing conditions |
| **Bit-width Error** | Signal width causing overflow/truncation |
    Incorrect sizing, indexing errors |
| **Simple Timing Error** | Basic signal timing adjustments | Wrong
    clock edge, missing register |
| **State Machine Error** | FSM definition or transition problems |
    Missing states, incorrect transitions |
| **Complex Timing Error** | Issues across combo/sequential circuits |
     CDC issues, handshaking failures |
| **Interface Error** | Missing/incorrect ports or protocols | Missing
     control signals, protocol violations |
| **Architectural Error** | Fundamental design structure problems |
    Performance bottlenecks, scalability issues |
| **Coding Pattern Error** | Non-synthesizable/non-standard practices
    | Inferred latches, combinational loops |
| **Cryptographic Error** | Security algorithm implementation issues |
    Side-channel vulnerabilities, weak randomization |
| **Software Error** | Embedded software code issues | Firmware bugs,
    driver interface problems |
| **Tool Error** | Issues causing tool failures | Simulator bugs,
    synthesis incompatibilities |
| **Test Error** | Testbench rather than design issues | Incorrect
    stimulus, missing checks |

### 5. Error Manifestation Analysis
Determine how the error was discovered:

| Manifestation | Description | Examples |
|---------------|-------------|----------|
| **Static Error** | Found during compilation/static analysis | Linter
    warnings, synthesis errors |
| **Oracle Error** | Simulation output mismatch | Waveform comparison
    failures |
| **Assertion Error** | Assertion check failure | Protocol violations,
    invariant failures |
| **Result Error** | Final result incorrect | Computation produces
    wrong output |
| **Timeout/Crash** | Simulation failure | Infinite loops, deadlocks |
| **CWE Match** | Matches known weakness catalog | Buffer overflows,
    authentication bypasses |
| **Performance Error** | Fails performance requirements | Timing
    closure failures, throughput issues |
| **Security Error** | Creates vulnerability | Side-channel leakage,
    privilege escalation |
| **Testbench Error** | Verification environment issue | Test stimulus
     errors, coverage problems |

### 6. Root Cause Analysis
- Conduct in-depth code logic analysis: Identify the specific line(s)
    of code where the error originated.
```

```
756    - Explain the faulty assumption, misunderstanding, or logic flaw that
757       led to the error.
758    - Examine discrepancies between architecture, specifications, and
759       implementation.
760    - Analyze interactions between modules, timing assumptions, or
761       misunderstandings of interfaces/protocols.
762    - Verify potential cross-module interaction issues.
763
764    ### 7. Fix Analysis
765    - Classify fix by scope (local/component/system), nature (corrective/
766       adaptive/perfective), and complexity
767    - Document specific changes and implementation strategy
768    - Verify fix validation methodology
769    - Check regression testing implementation
770    - Assess impact on performance, verification, and maintenance
771
772    ### 8. Test/Verification Assessment
773    - Identify verification methods used (simulation, formal, emulation)
774    - Document test outcomes before and after fix
775    - Evaluate coverage metrics and corner case handling
776    - Assess verification adequacy and potential improvements
777
778    ### 9. Dataset Suitability Evaluation
779    Evaluate based on these criteria:
780    - Is it a genuine RTL design error (clearly distinguish from testbench
781       , software, or tool errors based on the early filtering step)?
782    - Does the PR contain sufficient information for a comprehensive
783       analysis?
784    - Is the fix clear, focused, and educational for understanding the
785       error?
786    - Does it represent an important or common class of RTL errors?
787    - Does it add diversity (e.g., different error type, module type,
788       project context) to a potential error dataset?
789    State 'Yes' or 'No' and provide a concise justification.
790
791    ### 10. Comprehensive Report Generation
792    Generate the final report **strictly adhering** to the following
793       structure. The Classification Table must use the exact column
794       headers provided. If information for a section is unavailable in
795       the PR, explicitly state 'Information not available in PR'.
796
797    1.  **Executive Summary**: Concise overview of the PR, the identified
798       error, and the implemented fix.
799    2.  **Technical Analysis**: Detailed analysis incorporating findings
800       from Steps 1-8, using code snippets (pre/post fix) where relevant
801       to illustrate the error and fix.
802    3.  **Classification Table**:
803
804    | Error Type          | Error Manifestation | Root Cause
805                          | Fix Approach
806          | Fix Testing          | Dataset Recommendation    |
807    | ------------------ | ------------------ |
808       ------------------------------------------ |
809       ------------------------------------------ |
810       ---------------------- | --------------------- |
811    | [From Taxonomy]    | [From Taxonomy]    | [Detailed cause from Step
812       6]                  | [Specific changes & strategy from Step 7] | [
813       Methods from Step 8] | [Yes/No with reason] |
814
815    Example:
816    | Test Error          | Timeout/Crash       | Flawed error injection
817       method in test `chip_sw_lc_ctrl_program_error`: Forced error
818       signals instead of simulating illegal LC controller requests to
819       OTP controller, failing to test illegal LC state handling properly
820       . | Modified `create_illegal_lc_request_for_otp` function to
```

```
        inject errors by altering `lc_otp_program_i.state` to RAW (0),
        triggering proper OTP controller detection. | Verified `
        chip_sw_lc_ctrl_program_error` test now passes, showing OTP
        controller correctly detects/rejects illegal state. | No: Fix
        addresses a flaw in the test environment (error injection), not a
        functional error in the RTL design itself. |

## Guidelines for Quality Analysis
- **Be Thorough**: Analyze all available aspects of the PR, not just
    the most obvious issues.
- **Be Precise**: Use correct and consistent technical terminology (
    Verilog/SystemVerilog/VHDL, design concepts, verification terms).
- **Be Objective**: Base analysis on evidence found in the PR (code
    changes, descriptions, comments).
- **Be Educational**: Focus on extracting knowledge and insights
    beneficial to other hardware engineers.
- **Handle Ambiguity**: Note incomplete or unclear information and
    provide reasoned possible interpretations if necessary.
- **Focus on RTL**: Prioritize the analysis of hardware design and
    verification aspects over peripheral software or tooling issues,
    unless they are the primary subject of the PR fix.
- **Do Not Speculate**: Avoid guessing developer intent beyond what is
     evident in the PR and code.
- **Do Not Invent**: Do not add information not present in the PR
    materials.
- **Stay Focused**: Do not propose unrelated code improvements outside
     the scope of the identified error and its specific fix.

**IMPORTANT: Final Filtering Field**
After generating the complete analysis report as described above, you
    MUST append the following field at the absolute end of your
    response, based on your assessment in Step 9 (Dataset Suitability
    Evaluation):

`Filter_RTL_Error_Assessment: [Value]`

Where `[Value]` must be one of the following:
- `Relevant_RTL_Error`: If you answered 'Yes' to the suitability
    evaluation, indicating this PR represents a genuine, informative
    RTL design error fix suitable for a dataset.
- `Not_Relevant_RTL_Error`: If you answered 'No' to the suitability
    evaluation (e.g., it's a testbench error, software error,
    documentation change, tool issue, lacks information, or is
    otherwise unsuitable).

This field is crucial for automated filtering of the results. Ensure
    it is the very last line of your output.
```

## A.3  CASE STUDY: REPRODUCE ENVIRONMENT CONSTRUCTION ON OPENHWGROUP_CVA6-2035

The bugs from CVA6 in our proposed all falls in the category of logic error and specification implementation on the CSR part (Control State Registers). In RISC-V architecture, CSRs are critical for internal state control and interrupt and trap handling. Its specification is complex and verbose, making the design error-prone and hard to verify all edge cases. As a result, all reproducible hardware bugs in this repo is related to CSR.

However, many issues do not simply provide an automatic test for the bug, which we have to manually add during phase 4: manual validation and environment construction. Take pull request 2035 as an example, which fixes issue 1988. The issue states that "Previous privilege mode field, MPP, of mstatus or sstatus holds reserved value". To be specific, "mstatus" and "sstatus" are two CSRs in RISC-V holding the machine status in machine and supervisor privilege mode, respectively, and "MPP"(machine previous privilege) is a field (2 bits in a 32/64-bit register) in these two CSRs.

The privileged architecture specification of RISC-V specifies that if a reserved value or an unimplemented privilege level is written to the CSR, then it should become 0 when read.

As shown in Figure 6, an example scenario is provided, with an image of a waveform viewer showing the wrong internal state of the processor during simulation, where "PRIV_LVL_HS" is read from the "MPP" field. The processor in the scenario does not implement RVH extension, so the privilege level in MPP is unimplemented and should be reserved. Verifying by reading the waveform is difficult to automate. In open-source HDL repos, such cases are common obstacles for automatic data collections. Hardware developers are used to manual verification with tools like a waveform viewer, while weak issue instructions make bug reproduction difficult.

We read the issue and design a complete assembly code based on the RISCV-tests (RISC-V, 2025a) framework that triggers and detects the bug in software, and insert it into the test framework of the repo. The assembly ignores all irrelevant codes, directly loads the reserved value into the CSR, and reads it to a user register "x1". During testing, the assembly is loaded on the simulated CVA6 processor, whose user register states are compared with a standard behavioral model Spike (RISC-V, 2025b) to verify correctness. The bug will cause "x1" to be 2 instead of 0 after executing the "csrrci" instruction. To fix the bug, the patch modifies "core/csr_regfile.sv". An if statement should be inserted in two places to set the "MPP" field to 0 if RVH is not implemented, while "PRIV_LVL_HS" is written into the CSRs.

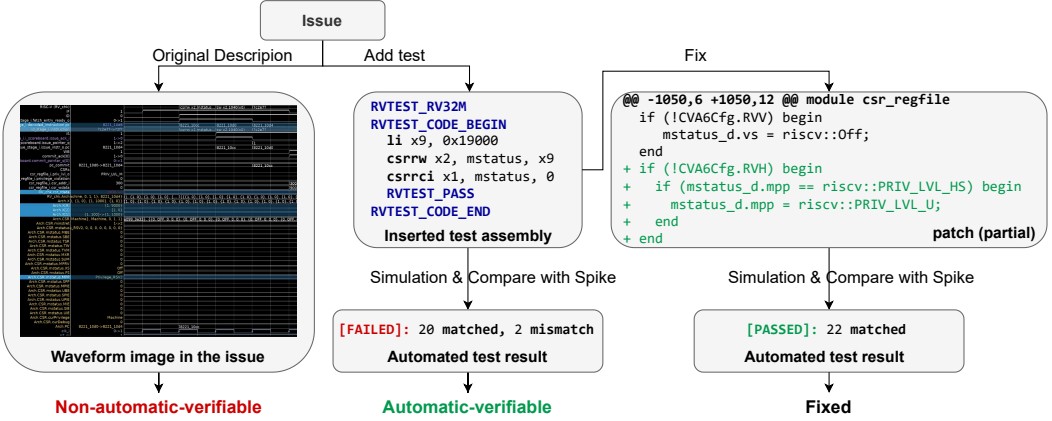

Figure 6: CVA6 example: inserting test and patch

## A.4    CASE STUDY: DIVERGENT AGENT TRAJECTORIES ON lowRISC__ibex-2261

We contrast the trajectories produced by GPT-5 and Claude-Sonnet-4 on the same issue: fixing the incorrect mtval value reported for illegal instructions in Ibex (GH PR #2261).

**Initial Context.**    The report describes that the co-simulation test prints mtval=0x00000000 even when the illegal instruction is 0x00000000, i.e. the controller masks the true faulting word. The golden fix adjusts the controller's illegal-instruction print path in rtl/ibex_controller.sv:175 so that compressed halfwords are sign-extended correctly.

**Golden Patch.**    The official fix touches a single file and replaces two lines in the controller (Listing 2). Whenever the decoder flags an illegal instruction, the controller now prints the actual 32-bit word: if the instruction was compressed, it concatenates the stored 16-bit halfword; otherwise it forwards the full instruction word. No other files are touched.

Listing 2: Golden patch for lowRISC__ibex-2261.

```diff
diff --git a/rtl/ibex_controller.sv b/rtl/ibex_controller.sv
index 157d05a6e..1b2666c0b 100644
--- a/rtl/ibex_controller.sv
+++ b/rtl/ibex_controller.sv
```

```
@@ −175,7 +175,7 @@ module ibex_controller #(
    if ((ctrl_fsm_cs == DECODE) && instr_valid_i
    && !instr_fetch_err_i
    && illegal_insn_d) begin
      $display("%t: Illegal instruction (hart %0x) at PC 0x%h: 0x%h", $time,
−   pc_id_i, id_stage_i.instr_rdata_i);
+   pc_id_i, instr_is_compressed_i ? {16'b0, instr_compressed_i} : instr_i );
    end
  end
```

**GPT-5 Trajectory (69 iterations).** GPT-5 performs a short sequence of `cat` and `rg` commands on the RTL hierarchy, quickly focusing on the fetch stage. It proposes the following patch (the only file touched):

```
diff −−git a/rtl/ibex_if_stage.sv b/rtl/ibex_if_stage.sv
@@ −512,7 +512,7 @@ module ibex_if_stage ...
−         instr_rdata_c_id_o    <= if_instr_rdata[15:0];
+         instr_rdata_c_id_o    <= if_instr_addr[1] ?
+                      if_instr_rdata[31:16] : if_instr_rdata[15:0];
@@ −526,7 +526,7
−         instr_rdata_c_id_o    <= if_instr_rdata[15:0];
+         instr_rdata_c_id_o    <= if_instr_addr[1] ?
+                      if_instr_rdata[31:16] : if_instr_rdata[15:0];
```

The proposal is incorrect (the bug lives in the controller, not in the IF stage), but the diff is clean: it modifies one RTL file, introduces no auxiliary artefacts, and keeps the repository pristine. GPT-5 therefore fails functionally yet exhibits disciplined engineering behaviour.

**Claude-Sonnet-4 Trajectory (100 iterations).** Claude-Sonnet-4 issues a much longer action sequence. After multiple failed runs of `Vibex_simple_system`, it begins copying host-side helpers and test programs into the tree, eventually installing an entire Verilator 4.210 distribution inside the repository. The final patch touches **93 files** (1.34 MB of diff). Besides a small change in `rtl/ibex_controller.sv`, the proposal rewrites the software test harness and commits large binary/third-party trees:

- `examples/sw/simple_system/common/common.mk` (switching the cross-compiler to `riscv64-unknown-elf`),
- a new C test (`test_illegal.c`) and modified `hello_test.c`,
- several thousand lines of `simple_system_cosim.log`,
- the whole `v4.210/share/...` Verilator install (man-pages, sample projects, headers, CMake build files, etc.).

Even though the RTL tweak is closer to the golden fix than GPT-5's attempt, the patch is unusable: review noise dwarfs the actual change, and committed third-party artefacts violate repository policy.

|  | GPT-5 | Claude-Sonnet-4 |
|---|---|---|
| Iterations | 69 | 100 (reaches cap) |
| Files touched | 1 | 93 |
| Diff size | 2 lines | 1.3 MB |
| Auxiliary artefacts | none | Verilator tree, logs, ad-hoc tests |
| Functional outcome | still incorrect | still incorrect |

Table 4: Trajectory comparison on `lowRISC_ibex-2261`.

**Summary.** This example illustrates the core behavioural gap: GPT-5 keeps the diff minimal—even when mis-localising the bug—while Claude-Sonnet-4's longer, noisier trajectory produces a patch

that is functionally suspect *and* engineering-wise unacceptable. The case study highlights why patch cleanliness, not just success rate, matters when assessing LLM agents in hardware repositories.

## A.5 Agent Scaffolding Configuration Details

We provide detailed documentation of our hardware-specific agent configurations for both SWE-Agent and OpenHands frameworks. This section addresses questions regarding (1) what information is included in each iteration's prompt, (2) what tools are available to the agents, and (3) what hardware-specific guidance is provided.

### A.5.1 Prompt Information Content Per Iteration

**Initial Prompt (First Iteration).**    The agent receives an initial prompt containing:

1. **Repository Context**: Full repository path and identification as a "SystemVerilog hardware repository."

2. **Issue Description**: Complete GitHub issue text including bug description, observed/expected behavior, reproduction steps (e.g., simulation commands), environment information (EDA tool versions, OS), code snippets, error messages, and commit SHA reference.

3. **Hardware-Specific Guidance**: Focus areas (RTL, DV components, supporting scripts), pre-installed toolchain notification (Verilator, Spike, VCS), environment activation commands, and simulator adaptation notes.

4. **Workflow Steps**: Repository exploration guidance for hardware directory structures (`hw/`, `dv/`, `util/`), failure reproduction via simulation/regression commands, minimal RTL/DV modification strategy, verification through re-running simulations, and corner case considerations (reset behavior, timing edges, X-propagation).

**Subsequent Iterations.**    Each subsequent iteration receives the full output from executed commands, including:

- **Compilation Errors**: Complete error messages from `verilator`, `vcs`, `gcc`, `make`, etc.
- **Simulation Logs**: Assertion failures, UVM error messages (`UVM_ERROR`, `UVM_FATAL`), and test transcripts.
- **File Contents**: Content of opened/read files (typically 100-line windows).
- **Current State**: Open file path and current working directory.

Notably, waveform data is provided only as text-based logs; VCD/FST files are not parsed visually. The model does *not* receive pre-computed textual diffs between buggy and fixed versions—it must generate patches via standard `git diff` operations.

### A.5.2 Available Tools and Action Space

**OpenHands (CodeActAgent) Tools.**    The CodeActAgent provides the following function-calling tools:

| Tool | Hardware Use Case |
| --- | --- |
| `execute_bash` | Execute shell commands: `make`, `vcs`, `verilator`, `dvsim`, `fusesoc` |
| `execute_ipython_cell` | Run Python code for script analysis, log parsing |
| `str_replace_editor` | Edit files with search/replace for RTL/DV source modification |
| `read_file` | Read file contents to examine RTL modules, testbenches |
| `finish` | Submit final solution and generate git patch |

Table 5: OpenHands tools available for hardware debugging tasks.

| Command | Signature | Hardware Use Case |
|---|---|---|
| `open` | `open <path> [line]` | Open `.sv`, `.v`, `.vh` files |
| `goto` | `goto <line>` | Navigate large RTL files |
| `scroll_down/up` | Navigate file window | Browse module implementations |
| `edit` | `edit <start>:<end>` | Modify RTL/DV code |
| `search_dir` | `search_dir <term> [dir]` | Find signal definitions, module instances |
| `search_file` | `search_file <term> [file]` | Locate specific logic in files |
| `find_file` | `find_file <name> [dir]` | Locate testbench/RTL files |
| `submit` | Submit patch | Generate `git diff` output |
| *Any bash* | Direct execution | `make`, `vcs`, simulation commands |

Table 6: SWE-Agent commands available for hardware debugging tasks.

**SWE-Agent Tools.** SWE-Agent provides shell-based commands for file navigation and editing:

No hardware-specific "oracles" are provided—the agent operates like a human engineer: searching code, editing SystemVerilog, invoking tools, and iterating based on logs.

### A.5.3 HARDWARE-SPECIFIC SYSTEM PROMPT

For OpenHands, we use a hardware-aware system prompt that differs from the standard Python/Java prompts. The key elements are shown in Listing 3.

Listing 3: Hardware-specific system prompt (OpenHands, abbreviated)

```
I've uploaded a SystemVerilog hardware repository in the directory
{workspace_dir_name}. Consider the following issue description:

<issue_description>
{instance.problem_statement}
</issue_description>

Can you help me implement the necessary RTL or testbench modifications
so that the requirements specified in the <issue_description> are met?

I've already taken care of changes to golden outputs or regression
manifests. Focus on RTL, DV components, or supporting scripts that
influence functional behaviour.

All hardware toolchains (Verilator, Spike, toolflows, Python packages)
are preinstalled inside a micromamba environment whose name matches
{workspace_dir_name}. Before running commands, execute:
  eval "$(/usr/bin/micromamba shell hook --shell=bash)" && \
  micromamba activate {workspace_dir_name}

Only Synopsys VCS is available as the commercial simulator. If the
issue mentions other simulators (Xcelium, Questa, etc.), adapt the
flow or scripts to use VCS instead.

Follow these steps:
1. Review the repository layout (hw/, dv/, util/) to understand the
   DUT and testbench structure.
2. Reproduce the reported failure by running the appropriate
   simulation or regression command (make/ninja, dvsim invocation).
3. Modify the relevant RTL or DV source files to fix the issue,
   aiming for the minimal change required.
4. Re-run the same simulation/regression to confirm the failure is
   resolved and that no new errors or assertions appear.
5. Think about corner cases such as reset behaviour, timing edges,
   or X-propagation. Add targeted checks or coverage if needed.
6. Compare your changes with the base commit {base_commit} to verify
   the requirements are fully addressed.
```

For SWE-Agent, we extend the framework to support language-specific instance templates. For Verilog/SystemVerilog, we provide a dedicated hardware-aware prompt that mirrors the guidance structure used in OpenHands. The hardware-specific instance template is shown in Listing 4.

Listing 4: SWE-Agent hardware-specific instance template for Verilog

```
We're currently solving the following issue within a SystemVerilog
hardware repository. Here's the issue text:
ISSUE:
{issue}

INSTRUCTIONS:
Now, you're going to solve this hardware bug on your own. Your
    terminal
session has started and you're in the repository's root directory.

HARDWARE REPOSITORY STRUCTURE:
- RTL source files are typically in `hw/`, `rtl/`, or `src/`
    directories
- Design verification (DV) and testbench files are in `dv/`, `tb/`, `
    test/`
- Build scripts and utilities are in `util/`, `scripts/`, or root
    directory

AVAILABLE TOOLS:
Tools of {language}: {language_specified_tools} could be used directly
    .
All hardware toolchains (Verilator, Spike, VCS) are preinstalled.

HARDWARE DEBUGGING WORKFLOW:
1. Explore the repository structure to understand DUT and testbench.
2. Reproduce the failure by running simulation/regression commands
    (e.g., make, ninja, dvsim.py). Capture and analyze the logs.
3. Identify root cause by examining RTL logic, signal propagation.
4. Modify RTL or DV source files with minimal changes required.
5. Re-run simulation to confirm fix and no new errors appear.
6. Consider hardware corner cases: reset behavior, clock domain
    crossings, timing edges, and X-propagation issues.

IMPORTANT TIPS:
1. Start by reproducing the bug using the simulation command in the
    issue.
2. Pay attention to signal widths, blocking vs non-blocking
    assignments.
3. Use the goto command to navigate large RTL files efficiently.
```

Both frameworks now provide comparable levels of hardware-specific guidance, enabling a fair comparison of their underlying agent architectures and tool-use capabilities.

### A.5.4 RUNTIME ENVIRONMENT CONFIGURATION

Per-repository Docker images include the following pre-installed toolchains:

- **Simulators**: Verilator 4.x+, Synopsys VCS (for OpenTitan DV flows)
- **RISC-V Tools**: Spike ISS, `riscv-tests` framework
- **Build Systems**: FuseSoC, Ninja, CMake, Make
- **Python Environment**: Python 3.10+ via micromamba with project dependencies

Environment variables are configured as follows:

```
export LANGUAGE=verilog
export USE_INSTANCE_IMAGE=true
export DOCKER_CLIENT_TIMEOUT=600   # Extended for large simulations
```

```
export COMPOSE_HTTP_TIMEOUT=600
```

### A.5.5  AGENT INTERACTION LOOP

The agent follows an iterative loop structure:

1. **Iteration 1**: Receives system prompt + issue description + hardware guidance; outputs thought + action (e.g., ls hw/ip/).

2. **Iterations 2–N**: Receives previous command output (compilation/simulation logs) + current file state + working directory; outputs thought + action (edit/run/search).

3. **Final Iteration**: Receives verification output (tests passing); outputs submit action to generate git patch.

Figure 7 illustrates this interaction loop schematically.

---

**Agent Iteration Loop**

**Iteration 1:**
 Input: System prompt + Issue description + HW guidance
 Output: Thought + Action (e.g., ls hw/ip/)

**Iteration 2–N:**
 Input: Previous command output (compilation/sim logs)
     + Current file state + Working directory
 Output: Thought + Action (edit/run/search)

**Final Iteration:**
 Input: Verification output (tests passing)
 Output: Submit action → generates git patch

---

Figure 7: Schematic of the agent interaction loop for hardware debugging.

### A.6  COST AND RUNTIME ANALYSIS

This section provides detailed cost and runtime statistics for each evaluated model on HDL-FixBench, addressing practical deployment considerations.

### A.6.1  API COST ANALYSIS

Table 7 reports the average token consumption and inference cost per task instance across all 57 benchmark cases. Costs are calculated based on each provider's published API pricing at the time of evaluation.

Table 7: API cost analysis per task instance on HDL-FixBench.

| Model | Avg. Input Tokens | Avg. Output Tokens | Avg. Cost/Case (USD) |
|---|---|---|---|
| GPT-5 | 3,375K | 74.7K | 1.67 |
| Claude-Sonnet-4 | 3,701K | 16.5K | 7.35 |
| Gemini-2.5-Pro | 5,860K | 51.3K | 2.30 |
| DeepSeek-V3.1 | 2,445K | 11.0K | 0.19 |
| Kimi-K2 | 3,838K | 19.7K | 2.35 |
| GLM-4.5 | 3,804K | 22.9K | 2.07 |
| Qwen3-Coder | 3,838K | 7.7K | 0.99 |

Two patterns emerge from the cost analysis. First, GPT-5 demonstrates strong cost-effectiveness: it achieves near-top Resolved rates at significantly lower cost than Claude-Sonnet-4 or Gemini-2.5-Pro, due to favorable pricing and efficient patch generation. Second, DeepSeek-V3.1 is extremely inexpensive (both due to lower pricing and shorter outputs) while achieving competitive performance. Claude-Sonnet-4, by contrast, is relatively expensive due to higher API pricing and the inability to exploit prompt caching in current agent frameworks.

### A.6.2 RUNTIME ANALYSIS

Table 8 reports end-to-end wall-clock times per task instance, including both LLM inference latency and hardware simulation execution time within the containerized environments.

Table 8: Runtime statistics per task instance on `HDL-FixBench`.

| Model | Mean (min) | Median (min) | Range (min) | Reasoning |
|---|---|---|---|---|
| GPT-5 | 24.2 | 16.6 | 1.5–105.3 | Yes |
| Claude-Sonnet-4 | 8.7 | 7.2 | 4.7–31.1 | No |
| Gemini-2.5-Pro | 17.8 | 15.0 | 3.4–41.0 | Yes |
| DeepSeek-V3.1 | 11.1 | 10.8 | 5.9–17.4 | No |
| Kimi-K2 | 15.2 | 15.0 | 6.9–20.6 | No |
| GLM-4.5 | 16.1 | 13.6 | 5.0–41.6 | Yes |
| Qwen3-Coder | 8.7 | 8.2 | 3.3–21.2 | No |

Notably, reasoning-enabled models do not uniformly outperform non-reasoning ones on `HDL-FixBench`. DeepSeek-V3.1 (non-reasoning) outperforms GLM-4.5 (reasoning) in both Resolved Rate and File-Level Precision, while Claude-Sonnet-4 (non-reasoning) achieves higher Resolved Rate than Gemini-2.5-Pro (reasoning). This suggests that for repository-level hardware repair, tool-use efficiency and cross-file navigation may be more critical than extended single-prompt reasoning.

### A.7 EDA TOOLCHAIN REQUIREMENTS

Table 9 summarizes the EDA toolchain requirements for each repository in `HDL-FixBench`. A key design goal was to maximize accessibility while preserving evaluation fidelity.

Table 9: EDA toolchain requirements by repository.

| Repository | Tasks | Simulator | Open-Source |
|---|---|---|---|
| Ibex | 7 | Verilator | Yes |
| CVA6 | 7 | Verilator | Yes |
| OpenTitan | 43 | Synopsys VCS | No |
| **Total** | **57** | — | 14 (24.6%) |

All 14 tasks from Ibex and CVA6 run entirely on Verilator and do not require any commercial simulator. These tasks use relatively minimal core/SoC testbenches with Spike co-simulation for RISC-V compliance checking.

For OpenTitan, the full DV/UVM regression environment is built around VCS-specific features and scripts. Achieving equivalent coverage with Verilator would require substantial re-engineering of the DV harness, which was not feasible within the scope of this work. We therefore mirror the upstream VCS-based flows to preserve evaluation fidelity.

Users with access to VCS can run the complete benchmark; others can still evaluate on the fully open-source Verilator subset (14 tasks).

### A.8 CROSS-MODEL VALIDATION OF PHASE-3 SEMANTIC FILTER

A potential concern is whether using Gemini 2.5 Pro as the Phase-3 semantic filter introduces bias favoring that model in subsequent evaluation. We address this with both design rationale and empirical cross-validation.

**Role of the Phase-3 Agent.** The agent performs binary, taxonomy-driven classification: does this PR correspond to a genuine RTL design error or a software-like/peripheral change? It does not rank difficulty or select "easy" PRs. Crucially, every PR labeled as RTL-relevant still undergoes Phase-4

manual validation—PRs that fail to yield a clean FAIL→PASS test are discarded. Misclassifications at Phase 3 affect recall (potentially missing some RTL bugs), but do not systematically select PRs based on repair difficulty or model-specific strengths.

**Cross-Model Agreement.** To test whether our filter choice induces architectural bias, we compared Phase-3 classification across three LLMs (Gemini-2.5-Pro, GPT-5, Kimi-K2) on samples from three repositories. Table 10 reports the results.

Table 10: Cross-model validation of Phase-3 semantic filter. "Sample": PRs entering Phase-3 (post-Phase-2 filtering). "Agreement": PRs where all three models gave identical labels. Model columns show PRs classified as RTL-relevant by each model.

| Repository | Sample | Agreement | Gemini | GPT-5 | Kimi-K2 |
|---|---|---|---|---|---|
| CVA6 | 100 | 84 | 86 | 78 | 82 |
| cv32e40p | 98 | 72 | 83 | 63 | 70 |
| OpenTitan | 100 | 85 | 27 | 21 | 22 |

Two conclusions emerge. First, semantic labeling is largely stable across quite different model architectures (72–85% three-way agreement). Second, where models differ, Gemini is consistently the most permissive—labeling the most borderline PRs as RTL-relevant. We chose Gemini deliberately for high recall, ensuring we do not discard potentially interesting RTL bugs before manual validation.

