# OpenReview forum: "HDL-FixBench: A Verifiable Repository-Level Benchmark for Hardware bug repair"
_ICLR.cc/2026/Conference — Submitted to ICLR 2026_

### Official Review · Reviewer_v2jx · 2025-10-31

**Soundness:** 2
**Presentation:** 2
**Contribution:** 1
**Rating:** 2
**Confidence:** 4

**Summary:**

This paper introduces HDL-FixBench, a benchmark that evaluates LLMs on repository-level hardware bug repair tasks. The contributions of the paper include: repository bug repair in real-world hardware projects, a distillation pipeline for hardware engineering tasks, and systematic evaluation of 7 LLMs across benchmarks.

**Strengths:**

* Repository-level hardware bug repair is important.
* Experiments on industrial projects demonstrate practical relevance.
* The work adapts software engineering benchmark practices for hardware-specific constraints.

**Weaknesses:**

* Since the authors target repository-level tasks, having only 57 instances from 3 repositories is quite limited.
* Excluding tasks >100 lines or >5 files makes me less excited, which contradicts the notion of "industrial-scale".
* The paper is missing ablation studies on filtering criteria, repository selection, or task complexity thresholds.
* Figure 4a is interesting but there is no discussion on the importance of different bug types.
* The only baseline is SWE-bench, which seems weak since this is a hardware benchmarking work. What about non-LLM baselines?

**Questions:**

Please see above.

---

> ### Author Response · Authors · 2025-11-26
> **Official Comment by Authors (Part1 / 3)**
>
> We appreciate the reviewer's detailed feedback and understand the concerns about benchmark scale and baseline comparisons. We address each point below.
>
> **W1 – Limited number of instances and repositories**
>
> We acknowledge that 57 instances from 3 repositories represents a focused rather than exhaustive benchmark. As detailed in our global response, this scale is largely dictated by the current HDL ecosystem and our requirement that each task be a real bug with a reproducible FAIL→PASS transition under the project's native regression flow. Starting from ≈30K PRs across fewer than ten Verilog/SystemVerilog repositories satisfying >500 PRs and >100 stars, our four-phase pipeline yields only 57 fully verifiable instances.
>
> Even at this size, HDL-FixBench exhibits clear discriminatory power: Resolved Rate spans 24–40% and File-Level Precision varies by over 30 points (≈41–77%) across models. The patch distribution also remains more complex than SWE-bench Verified (47% multi-file vs. 14%, 49% >20 lines vs. 18%). We view this release as a carefully curated foundation that we are actively extending.
>
> ---
>
> **W2 "Industrial-scale" vs. 100-line / 5-file thresholds**
>
> We appreciate the reviewer's concern and acknowledge that the relationship between our patch-size thresholds and the "industrial-scale" claim warrants clarification.
>
> **Clarifying scope and terminology.** HDL-FixBench focuses specifically on *functional bug repair*—tasks where a verifiable FAIL→PASS transition can be established. Large-scale refactorings, while important engineering activities, typically involve restructuring code for maintainability or performance without correcting a specific functional defect, making "correctness" difficult to define objectively. Accordingly, we use "industrial-scale" to describe the **source repositories** (OpenTitan, CVA6, Ibex), not the patch sizes: these are production-quality, industry-grade hardware projects with thousands of files, hundreds of thousands of lines of code, and comprehensive DV/UVM verification infrastructure (Fig. 4b). The thresholds constrain only the golden patch size, not the repository complexity or verification environment scope that agents must navigate.
>
> **Alignment with established practices and empirical evidence.** We agree that excluding large-scale modifications (>100 lines, >5 files) is a constraint, but this was a calibrated choice informed by both established practices and empirical observation. Patch-size filtering is standard in benchmark construction—SWE-bench Verified has over 98% of its patches naturally falling within our thresholds (Fig. 3). Crucially, even after filtering, HDL-FixBench retains a substantially more complex distribution: 47% of tasks require multi-file edits (vs. 14%) and 49% involve >20 lines changed (vs. 18%). The performance gap further confirms this complexity: the best model achieves only 40% Resolved on HDL-FixBench versus 70–75% on SWE-bench Verified.
>
> **Empirical analysis of larger-patch PRs.** We systematically analyzed PRs filtered out due to scale constraints. From approximately 1,700 PRs excluded by our thresholds in OpenTitan, Ibex, and CVA6, we examined over 200 candidates. The vast majority are code refactorings or feature additions rather than bug fixes. Among these, only ~5 PRs correspond to genuine bug fixes, of which only 2 could be successfully validated with reproducible FAIL→PASS transitions. Critically, all evaluated models achieved 0% Resolved on these two instances. This confirms that relaxing thresholds would primarily introduce invalid instances or uniformly unsolved tasks, neither of which contributes discriminative signal. We are actively incorporating the validated large-patch bug-fix cases as challenging extension instances to track future progress as model capabilities improve.

---

> > ### Author Response · Authors · 2025-11-26
> > **Official Comment by Authors (Part2 / 3)**
> >
> > **W3  Missing ablations on filtering criteria, repository selection, and thresholds**
> >
> > We appreciate the interest in understanding our pipeline's sensitivity. While systematic ablations are standard for model papers, benchmark construction has a different goal: ensuring task validity rather than optimizing a score. We provide transparency through detailed attrition statistics (Fig. 2) and address each aspect below.
> >
> > **Filtering criteria.** Our pipeline already exposes attrition at each stage: ~30K PRs → ~8K merged + issue-linked → ~500 after scale and semantic filtering → 57 after manual validation. The semantic filter's role is to ensure task validity, not to tune difficulty. Removing semantic filtering would retain documentation changes, CI tweaks, and testbench refactors that do not constitute RTL bugs—yielding an invalid benchmark rather than a different accuracy distribution.
> >
> > **Repository selection.** Unlike software benchmarks that draw from thousands of candidates, our search space is fundamentally constrained. Applying basic quality thresholds (>500 PRs, >100 stars, Verilog/SystemVerilog) to GitHub yields fewer than ten viable repositories. OpenTitan, Ibex, and CVA6 are essentially the only projects combining rich DV flows with enough genuine RTL bug-fix PRs to survive later phases. "Ablating" a repository here means losing a substantial fraction of the benchmark, not revealing a design choice among many alternatives.
> >
> > **Complexity thresholds.** We addressed this empirically: systematically analyzing PRs exceeding our thresholds shows most are feature additions or refactorings, and genuine large-patch bug fixes yield 0% Resolved across all models. Varying thresholds thus shifts the benchmark toward invalid or uniformly unsolved tasks rather than illuminating meaningful trade-offs.
> >
> >
> >
> > **W4 Figure 4a: importance of different bug types**
> >
> > The reviewer raises a good point. To clarify the importance of different bug types, we performed a per-category breakdown showing the number of resolved instances by category for each model.  The following table shows the best number of resolved instances by bug type for each model (maximum across both frameworks).
> >
> > | Model           | Logic (25) | Spec (12) | Interface (7) | TB/DV (6) | Timing (5) | Others (2) | Total |
> > | --------------- | ---------: | --------: | ------------: | --------: | ---------: | ---------: | ----: |
> > | GPT‑5           |          9 |         4 |             2 |         4 |          2 |          1 |    22 |
> > | Claude‑Sonnet‑4 |         10 |         4 |             3 |         3 |          2 |          1 |    23 |
> > | Gemini‑2.5‑Pro  |          6 |         3 |             3 |         2 |          2 |          0 |    16 |
> > | DeepSeek‑V3.1   |          7 |         4 |             3 |         3 |          2 |          0 |    19 |
> > | Kimi‑K2         |          8 |         4 |             1 |         0 |          2 |          0 |    15 |
> > | Qwen3‑Coder     |          7 |         3 |             2 |         2 |          2 |          0 |    16 |
> > | GLM‑4.5         |          5 |         4 |             2 |         1 |          2 |          0 |    14 |
> >
> > This analysis reveals the following insights:
> >
> > - **Logic / Spec bugs dominate in count but remain far from solved.**
> >     Even the strongest models resolve at most 10/25 Logic and 4/12 Spec bugs (≤40% and ≈33%), so HDL‑FixBench is not dominated by trivial logic fixes.
> >
> > - **Testbench/DV and Interface/Protocol bugs are key differentiators.**
> >     On Testbench/DV bugs, GPT‑5 resolves 4/6 while Kimi‑K2 resolves 0/6, with other models in between; on Interface/Protocol bugs, models resolve only 1–3 out of 7 cases. This indicates that understanding DV/UVM infrastructure and hardware protocols is a major axis along which models differ.
> >
> > - **Models have distinct capability profiles.**
> >     GPT‑5 is particularly strong on TB/DV bugs, Claude‑Sonnet‑4 leads on Logic bugs, Gemini‑2.5‑Pro and DeepSeek‑V3.1 are relatively better on Interface/Protocol bugs, while Kimi‑K2 is competitive on Logic/Spec but fails once DV code is involved. This category‑level view clarifies what kinds of hardware issues each model can or cannot handle, beyond the aggregate Resolved rate.
> >
> > **Action:** We will incorporate this analysis in the revised manuscript.

---

> > > ### Author Response · Authors · 2025-11-26
> > > **Official Comment by Authors (Part3 / 3)**
> > >
> > > **W5 Non‑LLM baselines (traditional RTL repair tools)**
> > >
> > > There is indeed a rich line of work on **automatic RTL repair**, for example, CirFix, RTL‑Repair, and FixRTL. These systems are powerful in their intended setting but have a fundamentally different interface and scale compared to HDL‑FixBench.
> > >
> > > - **Input modality mismatch.**
> > >     Tools such as CirFix and RTL‑Repair take as input a specific RTL module (or small design) plus a **failing testbench or formal property** that already reproduces the bug. They then search for patches that make this particular test (or set of tests) pass. In contrast, HDL‑FixBench tasks start from a *natural‑language issue description and a full repository*, mirroring how human engineers receive bug reports in practice. There is no pre‑identified module under repair, and the tests are woven into large DV/UVM flows rather than provided as an isolated harness.
> > > - **Scale mismatch.**
> > >     Existing RTL repair tools have been evaluated on designs at the level of individual modules or relatively small cores (hundreds to thousands of lines of code). CirFix, for instance, focuses on module‑level Verilog defects with targeted testbenches; RTL‑Repair’s evaluations involve a dozen real bugs on comparatively small designs. HDL‑FixBench, by design, operates on repositories like OpenTitan (≈7.5K files, ≈1.24M LoC) and CVA6/Ibex with full DV infrastructures (Fig. 4b).
> > > - **Task formulation.**
> > >     Traditional APR tools assume that failing tests and the boundary of the design under repair are *already given*, and they measure success by making those tests pass. HDL‑FixBench explicitly includes **bug localization, navigation, and DV tool use** as part of the task: agents must locate the relevant modules and tests, run the correct regressions, and then propose patches.
> > >
> > > Because of these fundamental modality and scale differences, we chose to focus HDL‑FixBench on comparing **LLM agents** under a standardized agentic framework (SWE‑Agent and OpenHands), and to use SWE‑bench Verified primarily as a *cross‑domain reference point* for difficulty rather than as a competing baseline.
> > >
> > > **References:**
> > >
> > > - Ahmad, Hammad, Yu Huang, and Westley Weimer. "CirFix: Automatically repairing defects in hardware design code." Proceedings of the 27th ACM International Conference on Architectural Support for Programming Languages and Operating Systems. 2022.
> > > - Laeufer, Kevin, et al. "Rtl-repair: Fast symbolic repair of hardware design code." Proceedings of the 29th ACM International Conference on Architectural Support for Programming Languages and Operating Systems, Volume 3. 2024.
> > > - Heidari, Mahsa, and Bijan Alizadeh. "FixRTL: Auto-correction of Multiple RTL Bugs by a New Feature Burst Clustering Algorithm and Mutation." ACM Transactions on Design Automation of Electronic Systems 30.4 (2025): 1-21.

---

### Official Review · Reviewer_8Fg3 · 2025-11-01

**Soundness:** 3
**Presentation:** 4
**Contribution:** 3
**Rating:** 8
**Confidence:** 4

**Summary:**

This paper introduces HDL-FixBench, a benchmark for evaluating LLMs on repository-level hardware bug repair tasks. The benchmark comprises 57 high quality instances from 3 open source hardware projects (OpenTitan, CVA6, Ibex), each with containerized EDA environments and carefully curated with human experts (demonstrated in Appendix A.3). The authors evaluate 7 state-of-the-art LLMs using SWE-Agent and OpenHands frameworks (questionable, though, whether these scaffoldings are appropriate for HW tasks), reporting ~40% success rate.

One interesting observation is that the paper doesn't just report numbers, it also “root causes” why hardware is so much harder by identifying critical weaknesses like the models' shallow understanding of hardware principles and their inability to coordinate multi-file edits across RTL, configuration, and verification components, thereby raising interesting questions about the need for superior long context and tool-use abilities for leading models.

**Strengths:**

- The benchmark fills an important void in hardware LLM evaluation, moving beyond component-level tasks to repository-scale challenges that better reflect real engineering workflows. The only comparable benchmark of this quality is CVDP Agentic.
- The multi-phase construction methodology is thorough. Particularly noteworthy is the extensive manual validation in Phase 4 (Section 3.1.4), where the authors manually verify each instance, create supplementary patches to resolve environment issues, and critically, write new test cases when automated tests don't exist. This level of human expertise ensures high-quality, verifiable tasks.
- The authors provide a base docker image with commercial tools and then open-source the entire pipeline for building upon it. This is a realistic compromise: acknowledges realities while maximize reproducibility.
- The use of an agentic filtering pipeline with GitHub MCP and Firecrawl tools to classify PRs against a hardware error taxonomy is scalable. This is a good contribution to the data scarcity problem in the field and could be replicated and applied to other repos.

**Weaknesses:**

- With only 57 instances from 3 repositories (all Verilog/SystemVerilog), the benchmark's generalizability is questionable, though benchmark creation process via human experts by nature is very challenging
- The paper mentions using SWE-Agent and OpenHands frameworks with "custom modifications" for Verilog/SystemVerilog, but doesn't provide sufficient detail about these agent configurations. What specific tools were made available to the agents? How were the action spaces defined? Were there hardware-specific prompts or guidance? In short, what are the agentic scaffolding used?
- The use of public repositories raises contamination concerns that aren't adequately addressed beyond noting "low success rates suggest memorization is insufficient."

**Questions:**

- Were hardware-specific tools (e.g., Verilator, synthesis tools) accessible during task execution?
- Did you use any hardware-specific prompting or guidance in the agent setup?
- Any human expert baselines on a subset of tasks?
- How many of the 57 tasks could potentially run with Verilator or Icarus Verilog?
- Which exact commercial EDA tools are required in your base Docker image? Are you using Synopsys VCS, Cadence Xcelium, etc?
- Could you provide a tool chain breakdown: X tasks need only basic simulation, Y need assertions, Z need full UVM?
- What was the runtime of your benchmark on standard inference hardware?

---

> ### Author Response · Authors · 2025-11-26
> **Official Comment by Authors (Part1 / 3)**
>
> We sincerely thank the reviewer for the positive and detailed review, and for highlighting both the strengths of HDL-FixBench and the important open questions around agent scaffolding and contamination. We address each concern below.
>
> ------
>
> **W1 Benchmark size and generalizability**
>
> We agree that 57 instances from 3 Verilog/SystemVerilog repositories limits statistical power and breadth. As detailed in the global response, this reflects (i) the intrinsic scarcity of large, actively maintained HDL repositories and (ii) our requirement that every task has a clean FAIL→PASS transition under native regression flows. Starting from ≈30K PRs across fewer than ten eligible repositories, the four-phase pipeline distills down to 57 fully reproducible instances.
>
> Despite this size, HDL-FixBench shows strong discriminative power: models span 24–40% Resolved Rate, File-Level Precision varies by >30 points, and patch distributions remain more multi-file and higher-line-count than SWE-bench Verified (Fig. 3). We view this version as a carefully curated, high-fidelity foundation that we are actively extending.
>
> ------
>
> **W2 Agent scaffolding details**
>
> We provide a detailed breakdown of our hardware-specific configurations:
>
> **Tools and action space.** Both frameworks retain the standard "edit file + run command" paradigm. The agent can:
>
> - Execute shell commands to run `make`, `verilator`, `VCS`, `dvsim`/`fusesoc`, and RISC-V simulators (Spike)
> - View/edit RTL and DV code via file read/write tools
> - Search directories and files to locate signal definitions, module instantiations, and test entry points
>
> No hardware-specific "oracles" are provided—the agent operates like a human engineer: searching code, editing SystemVerilog, invoking tools, and iterating based on logs.
>
> **System prompts.** The prompt explicitly:
>
> - Identifies the repository as a SystemVerilog hardware project
> - Specifies that fixes should modify RTL/DV logic, not mask errors via golden outputs or test lists
> - Suggests a debugging workflow: understand repo structure → reproduce failure → make minimal changes → re-run simulation → consider hardware corner cases (reset, timing, X-propagation)
>
> **Runtime environment.** Per-repository Docker images include Verilator, Spike, Synopsys VCS, FuseSoC, Ninja/CMake, and Python via micromamba. If the issue references other commercial simulators (Xcelium/Questa), scripts are adapted to VCS for consistency.
>
> **Action:** We will include configuration details in the supplementary materials.
>
> ---
>
> **W3 Contamination and public‑data training corpora**
>
> We agree that some contamination is possible with public GitHub data. Our goal is to argue that results are unlikely to be dominated by patch memorization.
>
> **Corpus-level evidence.** Verilog/SystemVerilog forms a tiny fraction of typical training corpora (The Stack v2). Even if parts of our source repositories appear in training sets, they constitute a much smaller signal than popular Python/C++ projects.
>
> **Behavioral evidence.** If memorization were dominant, we would expect high Resolved rates and precise diffs. Instead, success rates remain modest (best 40.3%), agents require dozens of tool-use steps, and patches are often larger and noisier than ground truth. Failure analysis (Sec. 4.3) reveals typical reasoning errors—mis-localization, missing cross-file updates, hardware invariant violations—rather than verbatim reproduction.
>
> **Direct probe.** Following Liang et al. (2025), we tested whether models can predict the buggy file path from the issue description alone (no repository context). On HDL-FixBench, accuracies range from 29.8%–47.4% (GPT-5: 47.4%, Claude-Sonnet-4: 45.6%, Gemini-2.5-Pro: 33.3%). This is well below the ~76% Liang et al. report for SWE-bench Verified and even below their ~53% "outside-repo" control group, suggesting limited memorization of our benchmark instances. A mild positive correlation with overall performance (GPT-5 /Claude-Sonnet-4 > Gemini-2.5-Pro) is consistent with project-level familiarity rather than exact solution lookup.
>
> Taken together, these points suggest that HDL‑FixBench may share the usual contamination caveats of public‑code benchmarks, but that the scores we report mainly reflect models’ ability (or inability) to solve difficult, multi‑step hardware debugging tasks, rather than being driven by large‑scale memorization of past patches.
>
> **Reference**
>
> - Liang, Shanchao, Spandan Garg, and Roshanak Zilouchian Moghaddam. "The SWE-Bench Illusion: When State-of-the-Art LLMs Remember Instead of Reason." *arXiv preprint arXiv:2506.12286* (2025).

---

> ### Author Response · Authors · 2025-11-26
> **Official Comment by Authors (Part2 / 3)**
>
> **Q1. Were hardware‑specific tools accessible during execution?**
>
> Yes. All hardware‑specific tools used in the source projects are installed in the per‑PR Docker images and accessible via the shell tool. Concretely:
>
> - **CVA6 & Ibex (14 tasks)**: Verilator‑based simulations, plus project‑specific harnesses and Spike/`riscv-tests` where needed (e.g., the CSR case study).
> - **OpenTitan (43 tasks)**: VCS‑based DV/UVM regressions invoked through `dvsim.py` and Make/Ninja; Verilator does not yet fully support these DV flows, so we mirror the upstream setup.
>
> The agent calls these tools via standard commands (e.g., `dvsim.py`, `make`, `verilator`, `vcs`), and their logs enter the context as plain text.
>
> ------
>
> **Q2. Did you use hardware‑specific prompting or guidance?**
>
> Yes, we do use lightweight, uniform hardware-specific prompting, but it is intentionally high-level and does not encode any instance-specific hints or bug solutions. We use a hardware‑aware system prompt that:
>
> - identifies the project as a Verilog/SystemVerilog hardware repository,
> - explains the typical directory structure (`hw/`, `rtl/`, `dv/`, `test/`, `util/`),
> - lists available simulators and how to activate the micromamba environment, and
> - suggests a hardware‑specific debugging workflow (reproduce failing regression, inspect RTL + DV, consider reset/X‑propagation/timing, apply minimal patch, re‑run same regression).
>
> We do **not** provide any privileged hints about the location of the fix or a pre‑selected file list; navigation still happens via `ls`, `grep`, etc.
>
> ---
>
> **Q3. Any human expert baselines?**
>
> We did not run a controlled human-expert benchmark due to limited verification engineer availability. From Phase-4 environment construction experience—where we had to understand each issue, bring up regression flows, and sometimes design new tests—our informal impression is that an expert starting from the same inputs as the agent would typically need tens of minutes to hours for diagnosis and clean fix. A proper human study would be valuable future work but is outside the scope of this release.
>
> ------
>
> **Q4. How many of the 57 tasks can run with Verilator or Icarus Verilog?**
>
> All **14 tasks from Ibex and CVA6** run entirely on **Verilator** in our setup and do not require any commercial simulator. We do not currently use Icarus Verilog for any task.
>
> For **OpenTitan**, we attempted to rely on open‑source flows where practical, but the full DV/UVM regression environment is built around VCS‑specific features and scripts. Achieving equivalent coverage and behavior with Verilator would require substantial re‑engineering of the DV harness and was not feasible within the scope of this work. Thus, for these 43 tasks we use the same VCS‑based flows as the upstream project to preserve fidelity.
>
> ------
>
> **Q5. Which commercial EDA tools are required?**
>
> The only commercial simulator we rely on is **Synopsys VCS**, used for the OpenTitan tasks through its DV/UVM regressions. CVA6 and Ibex tasks run on Verilator and do not depend on commercial simulators.
>
> The underlying Docker build recipes follow the same pattern described in Sec. 3.1.4: a base image with the licensed tools, plus repository‑specific layers and per‑instance patches and test scripts. Users with access to VCS can recreate compatible images; others can still run the Verilator‑only subset.

---

> > ### Author Response · Authors · 2025-11-26
> > **Official Comment by Authors (Part3 / 3)**
> >
> > **Q6. Toolchain breakdown: basic simulation vs DV/UVM/assertions**
> >
> > Roughly:
> >
> > - **Ibex tasks (7)**: “basic simulation only” — relatively minimal core/SoC testbench, no full DV/UVM environment.
> > - **CVA6 + OpenTitan tasks (50)**: exercised via the projects’ standard DV/UVM flows, with integrated assertions and checkers. Tasks that rely on assertions are part of these full DV flows rather than a separate “assertion‑only” chain.
> >
> > So the majority of HDL‑FixBench (50/57) involves working within industrial‑style DV/UVM infrastructures rather than minimal unit tests.
> >
> > ------
> >
> > **Q7. Runtime of the benchmark**
> >
> > End-to-end wall-clock times per instance (including tool calls):
> >
> > | Model                      | Mean (min) | Median (min) | Range (min) |
> > | -------------------------- | ---------- | ------------ | ----------- |
> > | Claude-Sonnet-4            | 8.7        | 7.2          | 4.7–31.1    |
> > | Qwen3-Coder                | 8.7        | 8.2          | 3.3–21.2    |
> > | DeepSeek-V3.1              | 11.1       | 10.8         | 5.9–17.4    |
> > | Kimi-K2                    | 15.2       | 15.0         | 6.9–20.6    |
> > | GLM-4.5 (reasoning)        | 16.1       | 13.6         | 5.0–41.6    |
> > | Gemini-2.5-Pro (reasoning) | 17.8       | 15.0         | 3.4–41.0    |
> > | GPT-5 (reasoning)          | 24.2       | 16.6         | 1.5–105.3   |
> >
> > These numbers combine LLM inference time and non‑trivial hardware regression time inside the containers. Explicit reasoning models tend to generate longer outputs and run more iterations, which increases latency, but they do not always translate that extra computation into higher Resolved rates—an interesting efficiency–performance trade‑off revealed by HDL‑FixBench.

---

### Official Review · Reviewer_suzP · 2025-11-01

**Soundness:** 2
**Presentation:** 2
**Contribution:** 2
**Rating:** 2
**Confidence:** 4

**Summary:**

This paper introduces HDL-FixBench, the first benchmark for evaluating the repair of repository-level (RTL) hardware bugs using LLMs. The authors present a screening and build process for evaluating 57 real-world bug instances collected from three industrial-grade open-source hardware projects (OpenTitan, CVA6, Ibex).

**Strengths:**

This paper is the first benchmark focusing on repository-level hardware bug fixing. The multi-stage build process is a crucial aspect of this paper, particularly the use of an LLM Agent for semantic filtering (stage 3) to extract "real RTL bugs" and the manual verification in stage 4, which demonstrates a high degree of rigor.

**Weaknesses:**

1. The current number of 57 instances from 3 projects is relatively small. While we understand the scarcity of data in this field, the small scale may limit the diversity of benchmarks and the statistical significance when evaluating the models' capabilities.
2. Using Gemini 2.5 Pro as an agent to filter PRs may introduce bias, leading to unfair subsequent evaluations.
3. The author intentionally excluded tasks involving changes to more than 100 lines or 5 files. While this helps make the problem "manageable," it also arbitrarily excludes the many more complex refactorings and system-level bugs that exist in the real world. This makes the benchmark only representative of "small-scale, localized" hardware fixes, thus diminishing its claimed "repository-level" value.

**Questions:**

1. Does the benchmark's "resolved rate" depend solely on passing the testbench? Is it possible for a patch to pass the tests while simultaneously being completely unacceptable in practice (due to the introduction of numerous irrelevant files)? Does the highest score of 40.3% include this scenario?
2. How can we ensure that using Gemini 2.5 Pro as a filter does not bias the benchmark set towards tasks that favor the model architecture? Has cross-validation been performed, for example, using Claude-Sonnet-4 as a filter to see if it selects 57 different tasks?

---

> ### Author Response · Authors · 2025-11-26
> **Official Comment by Authors (Part1 / 2)**
>
> We thank the reviewer for recognizing the "high degree of rigor" in our multi-stage curation process, particularly the manual verification in Phase 4. We address the concerns about dataset size, filtering bias, and "repository-level" scope below.
>
> ------
>
> **W1 Benchmark size and statistical robustness**
>
> We refer to our global response for detailed discussion. In brief, the 57-instance scale reflects fundamental HDL ecosystem scarcity: starting from ~30K PRs across fewer than ten eligible repositories, only 57 can be turned into containerized FAIL→PASS tasks after our four-phase pipeline. Despite this size, the benchmark exhibits clear discriminative power: Resolved rates span 24–40%, and File-Level Precision varies by >30 points across models (Table 2). We position this as a carefully curated foundation that we are actively extending.
>
> ------
>
> **W2/Q2 Bias from using Gemini 2.5 Pro as the semantic PR filter**
>
> This is an important methodological question. We address it with both design rationale and empirical cross-validation.
>
> **Role of the Phase-3 agent.** The agent performs binary, taxonomy-driven classification: does this PR correspond to a *genuine RTL design error* or a software-like/peripheral change? It does not rank difficulty or select "easy" PRs. Crucially, every PR labeled as RTL-relevant still undergoes Phase-4 manual validation—PRs that fail to yield a clean FAIL→PASS test are discarded. Misclassifications at Phase 3 affect *recall* (potentially missing some RTL bugs), but do not systematically select PRs based on repair difficulty or model-specific strengths.
>
> **Cross-model validation.** To test whether our filter choice induces architectural bias, we compared Phase-3 classification across three LLMs (Gemini-2.5-Pro, GPT-5, Kimi-K2) on samples from three repositories. We sampled PRs that passed Phase-2 filters (merged + linked to issue) to ensure the comparison reflects the actual candidate pool entering Phase-3.
>
> | Repository | Sample | Agreement | Gemini | GPT-5 | Kimi-K2 |
> | ---------- | ------ | --------- | ------ | ----- | ------- |
> | CVA6       | 100    | 84        | 86     | 78    | 82      |
> | cv32e40p   | 98     | 72        | 83     | 63    | 70      |
> | OpenTitan  | 100    | 85        | 27     | 21    | 22      |
>
> > All values are counts. "Sample": PRs entering Phase-3 (post-Phase-2 filtering). "Agreement": PRs where all three models gave identical labels (all RTL-relevant or all non-RTL-relevant). "Gemini/GPT-5/Kimi-K2": PRs classified as RTL-relevant by each model. Higher counts indicate a more permissive filter.
>
> The lower RTL-relevant ratio in OpenTitan reflects the project's nature: as a full SoC platform, the majority of its PRs address firmware, tooling, documentation, and CI infrastructure rather than RTL design bugs—exactly the software-like changes our filter is designed to exclude.
>
> Two conclusions emerge: (i) **semantic labeling is largely stable** across quite different models (72–85% three-way agreement), and (ii) where they differ, **Gemini is consistently the most permissive**—labeling the most borderline PRs as RTL-relevant. We chose Gemini deliberately for **high recall**, ensuring we do not discard potentially interesting RTL bugs before manual validation.
>
> **On potential evaluation bias.** If the filter systematically biased the dataset toward Gemini-favorable tasks, we might expect Gemini-2.5-Pro to excel on HDL-FixBench. In practice, Gemini has the lowest Resolved Rate among proprietary models (28.1% vs. 38.6% GPT-5, 40.3% Claude-Sonnet-4) and mediocre File-Level Precision (~50%). This does not rule out all forms of bias, but provides some reassurance that using Gemini as the Phase-3 filter does not confer a systematic advantage to that model in evaluation.

---

> ### Author Response · Authors · 2025-11-26
> **Official Comment by Authors (Part2 / 2)**
>
> **W3 “Repository‑level” vs. small patches**
>
> We appreciate the reviewer raising this important point, and we acknowledge that the patch-size thresholds represent a deliberate design choice that warrants careful justification.
>
> **Clarifying scope and terminology.** HDL-FixBench focuses specifically on *functional bug repair*—tasks where a verifiable FAIL→PASS transition can be established. Large-scale refactorings, while important engineering activities, typically involve restructuring code for maintainability or performance without correcting a specific functional defect, making "correctness" difficult to define objectively. Accordingly, "repository-level" refers to the **context and evaluation mode**, not patch size: agents operate on the entire repository (thousands of files, hundreds of thousands of lines—Fig. 4b) with no pre-selected file list or fault localization hints. A one-line clock-domain crossing fix can require hours of expert analysis to identify within such a massive codebase.
>
> **Alignment with established practices and empirical evidence.** We agree that excluding large-scale modifications (>100 lines, >5 files) is a constraint, but this was a calibrated choice informed by both established practices and empirical observation. Patch-size filtering is standard in benchmark construction, SWE-bench Verified has over 98% of its patches naturally falling within our thresholds (Fig. 3). Crucially, even after filtering, HDL-FixBench retains a substantially more complex distribution: 47% of tasks require multi-file edits (vs. 14%) and 49% involve >20 lines changed (vs. 18%). The performance gap further confirms this complexity: the best model achieves only 40% Resolved on HDL-FixBench versus 70–75% on SWE-bench Verified.
>
> **Empirical analysis of larger-patch PRs.** We systematically analyzed PRs filtered out due to scale constraints. From approximately 1,700 PRs excluded by our thresholds in OpenTitan, Ibex and CVA6, we examined over 200 candidates. The vast majority are code refactorings or feature additions rather than bug fixes. Among these, only ~5 PRs correspond to genuine bug fixes, of which only 2 could be successfully validated with reproducible FAIL→PASS transitions. Critically, all evaluated models achieved 0% Resolved on these two instances. This confirms that relaxing thresholds would primarily introduce invalid instances or uniformly unsolved tasks, neither of which contributes discriminative signal. We are actively incorporating the validated large-patch bug-fix cases as challenging extension instances to track future progress as model capabilities improve.
>
> ------
>
> **Q1. Does the Resolved Rate depend solely on passing the testbench? Are “unacceptable” patches counted?**
>
> Yes, by definition, Resolved depends solely on the test script returning PASS, so it can count engineering-unacceptable patches; this is precisely why we introduce File-Level Precision as a complementary metric.
>
> On HDL-FixBench, this distinction is crucial. Claude-Sonnet-4 achieves the highest Resolved (40.3%) but low File-Level Precision (\~47%), whereas GPT-5 has slightly lower Resolved (38.6%) but much higher Precision (\~77%). The Ibex-2261 case study (Appendix A.4) illustrates this vividly: Claude produces a 93 files,1.3 MB diff including third-party code and logs, engineering-wise unacceptable.
>
> The 40.3% top Resolved Rate **does** include such noisy patches. We view this as an important finding that motivates our dual-metric evaluation, and we explicitly discuss it as a limitation of purely test-based metrics.

---

### Official Review · Reviewer_Juiu · 2025-11-01

**Soundness:** 3
**Presentation:** 3
**Contribution:** 3
**Rating:** 6
**Confidence:** 5

**Summary:**

This paper introduces HDL-FixBench, a new benchmark designed to evaluate large language models (LLMs) on their ability to automatically detect and fix functional and syntactic errors in hardware description language (HDL) code. The benchmark is built from three open-source CPU core repositories, with each bug instance accompanied by the corresponding fixed version verified through simulation. The benchmark provides a well-defined evaluation framework including metrics such as resolution rate and file-level precision, ensuring that both syntactic correctness and functional validity are assessed. The authors systematically evaluate seven different LLMs, including both open-source and proprietary models, using a uniform multi-turn prompt-based setup. They analyze success rates across repair complexity, bug type, and interaction depth, highlighting key challenges in applying LLMs to HDL repair.

**Strengths:**

1. The paper addresses a timely and underexplored problem -- automated code repair in the HDL domain -- by introducing a benchmark specifically tailored for hardware design.
2. The work presents a comprehensive and methodical evaluation of seven different LLMs (including both base and reasoning-enabled variants), measuring resolution rate, functional correctness, and file-level precision.
3. The authors offer valuable insights on how HDL repair differs from traditional software engineering tasks.

**Weaknesses:**

1. The benchmark size is small (57 instances), which limits statistical robustness and the ability to draw strong generalization conclusions.
2. The dataset's representativeness is narrow, as all three source repositories correspond to CPU core designs. This design bias restricts the benchmark's ability to test model adaptability to other hardware categories such as DSP modules, CNN accelerators, or LLM hardware implementations.

**Questions:**

1. Please clarify what specific information is included in each iteration's prompt—for example, does the model receive compilation errors, waveform logs, or only textual diffs between buggy and fixed versions? Such details are important for interpreting how much context the model relies on for effective debugging.
2. Please provide the API cost for each model when running the whole benchmark.
3. Do you enable reasoning mode for those proprietary models? Any insights on the reasoning time or comparing with the non-reasoning models?
4. Can you provide more diverse designs like CNN or LLM accelerators written in RTL?

Nit: Please use citep for the inlined citations.

---

> ### Author Response · Authors · 2025-11-26
> **Official Comment by Authors (Part1 / 2)**
>
> We thank the reviewer for recognizing HDL-FixBench as addressing a "timely and underexplored problem" and for appreciating the "comprehensive and methodical evaluation" across multiple models. We also value the constructive suggestions on expanding benchmark diversity. We address the specific concerns below.
>
> **W1 – Benchmark size (57 instances).** We refer the reviewer to our global response for a detailed discussion of ecosystem-level HDL scarcity. In brief: starting from ~30K PRs across three industrial-grade repositories, our four-phase pipeline yields only 57 containerized FAIL→PASS tasks. This reflects both the fundamental data scarcity in open-source HDL and our requirement that every task be a real bug with execution-grounded verification.
>
> **W2 & Q4 – Repository diversity (all three are CPU cores).** We agree that broader design coverage would strengthen HDL-FixBench. Our initial selection is driven by two constraints: (i) OpenTitan, Ibex, and CVA6 are among the very few Verilog/SystemVerilog projects satisfying >500 PRs and >100 stars, and (ii) they provide actively maintained verification flows with enough genuine RTL bug-fix PRs to survive our pipeline.
>
> To probe whether accelerator-style repositories could be included, we ran a pilot on CFU-Playground (an open-source CFU accelerator framework). Notably, CFU-Playground is among the most active accelerator repositories we could identify (>500 stars, >500 PRs)—most other accelerator projects have far less community activity. Starting from ~500 PRs, only 17 remain after filtering to merged PRs linked to issues that touch RTL files. Even under relaxed thresholds (≤500 lines, ≤20 files), only 4 PRs remain—all feature additions or refactorings rather than bug fixes. This mirrors what we observe in other non-CPU repositories: technically interesting designs, but extremely few bug-fix PRs convertible into executable tasks.
>
> We also note that CPU cores are arguably among the most representative hardware designs for evaluating bug repair capabilities. CPU designs are **control-logic-intensive**: they feature complex finite state machines, intricate pipeline control, exception handling, and multi-module coordination—exactly the scenarios where subtle bugs arise and require deep reasoning to fix. In contrast, accelerators (e.g., for CNN or matrix operations) tend to be more **dataflow-dominated**, with relatively regular, repetitive structures and fewer control-path edge cases. The bug categories in HDL-FixBench (logic errors, FSM issues, interface/protocol violations, timing/synchronization bugs) reflect challenges that generalize across hardware domains, even though the source repositories are CPU-centric.
>
> We are actively extending the benchmark to additional candidates (other CORE-V cores, Caliptra-RTL, Black-Parrot, bsg_manycore). Early experiments confirm the pipeline generalizes, but the underlying PR scarcity remains the bottleneck. We view the current 57-instance benchmark as a carefully curated foundation that we are actively extending.

---

> > ### Author Response · Authors · 2025-11-26
> > **Official Comment by Authors (Part2 / 2)**
> >
> > **Q1 – What information is included in each iteration’s prompt?**
> > Our setup is an interactive debugging loop. In the *first* iteration, the agent receives: (1) a system prompt describing the repository as a SystemVerilog hardware project with guidance on the RTL/DV tree layout and available toolchains, and (2) the full GitHub issue text (bug description, expected vs. observed behavior, reproduction commands, error messages, and any code snippets or waveform-related artifacts).
> >
> > From the *second* iteration onward, the agent sees the full stdout/stderr of all executed commands: compiler/elaboration errors, simulation transcripts, assertion failures, UVM messages, and file contents it chose to open. We do **not** provide pre-computed diffs between buggy and fixed versions—the agent generates patches based on observed errors. Waveforms are not parsed directly, but any waveform-related textual output in logs is visible.
> >
> > **Action:** We will clarify these details in the revised manuscript.
> >
> > ------
> >
> > **Q2 – API cost for each model on the whole benchmark.**
> >
> > | Model           | Avg. Input Tokens | Avg. Output Tokens | Avg. Cost/Case (USD) |
> > | --------------- | ----------------- | ------------------ | -------------------- |
> > | GPT-5           | 3,375K            | 74.7K              | 1.67                 |
> > | Claude-Sonnet-4 | 3,701K            | 16.5K              | 7.35                 |
> > | Gemini-2.5-Pro  | 5,860K            | 51.3K              | 2.30                 |
> > | DeepSeek-V3.1   | 2,445K            | 11.0K              | 0.19                 |
> > | Kimi-K2         | 3,838K            | 19.7K              | 2.35                 |
> > | GLM-4.5         | 3,804K            | 22.9K              | 2.07                 |
> > | Qwen3-Coder     | 3,838K            | 7.7K               | 0.99                 |
> >
> > Two patterns stand out. First, GPT-5 demonstrates strong cost-effectiveness: it achieves near-top Resolved rates at significantly lower cost than Claude-Sonnet-4 or Gemini-2.5-Pro, due to favorable pricing and efficient patch generation. Second, DeepSeek-V3.1 is extremely inexpensive (both due to lower pricing and shorter outputs) while achieving competitive performance. Claude-Sonnet-4, by contrast, is relatively expensive due to higher API pricing and the inability to exploit prompt caching in current agent frameworks.
> >
> > **Action:** We will add this cost analysis to the revised manuscript.
> >
> > ------
> >
> > **Q3 – Reasoning modes, latency, and comparison with non-reasoning models.**
> >
> > We enabled reasoning modes for GPT-5, Gemini-2.5-Pro, and GLM-4.5. For Claude-Sonnet-4, we used regular tool-use mode (without explicit "thinking") to match the SWE-bench Verified configuration for fair comparison.
> >
> > | Model           | Mean Runtime (min) | Median Runtime (min) | Reasoning Mode |
> > | --------------- | ------------------ | -------------------- | -------------- |
> > | GPT-5           | 24.2               | 16.6                 | Yes            |
> > | Claude-Sonnet-4 | 8.7                | 7.2                  | No             |
> > | Gemini-2.5-Pro  | 17.8               | 15.0                 | Yes            |
> > | DeepSeek-V3.1   | 11.1               | 10.8                 | No             |
> > | Kimi-K2         | 15.2               | 15.0                 | No             |
> > | GLM-4.5         | 16.1               | 13.6                 | Yes            |
> > | Qwen3-Coder     | 8.7                | 8.2                  | No             |
> >
> > Reasoning-enabled models tend to have longer runtimes primarily because they output substantially more tokens for chain-of-thought traces. This is evident from the Q2 data: GPT-5 outputs 74.7K tokens on average versus 16.5K for Claude-Sonnet-4 and 11.0K for DeepSeek-V3.1.
> >
> > Importantly, reasoning-enabled models do not uniformly outperform non-reasoning ones on this benchmark. DeepSeek-V3.1 (non-reasoning) outperforms GLM-4.5 (reasoning) in both Resolved Rate (33.3% vs. 24.6%) and File-Level Precision (73.7% vs. 54.9%), while Claude-Sonnet-4 (non-reasoning) achieves higher Resolved Rate than Gemini-2.5-Pro (reasoning) at 40.3% vs. 28.1%. This suggests that for repository-level hardware repair, **tool-use efficiency and cross-file navigation** may matter more than extended single-prompt reasoning.
> >
> > ------
> >
> > **Nit – Citation style.**
> >
> > **Action:** Thank you for pointing this out. We will fix it in the revised manuscript.

---

> > > ### Comment · Reviewer_Juiu · 2025-11-28
> > >
> > > I appreciate the authors' efforts in clarifying my questions. The responses were helpful for better understanding the proposed techniques and results. I also recognize the fundamental limitations in current HDL dataset curation. However, I still believe the authors should make a stronger attempt to expand the size and diversity of the benchmark. RISC-V cores or other diverse designs written in DSLs that can be converted to HDL may serve as good candidates. I will therefore maintain my current score.

---

> > > > ### Author Response · Authors · 2025-12-04
> > > >
> > > > We sincerely thank the reviewer for the continued engagement and constructive suggestions.
> > > >
> > > > We fully agree that expanding the benchmark's size and diversity is an important direction. The reviewer's suggestion regarding DSL-based designs (e.g., Chisel) is well-aligned with our revised limitations section, where we explicitly mention Chisel support as a planned extension.
> > > >
> > > > We are actively exploring Chisel-based repositories such as rocket-chip and chipyard. However, integrating these presents additional technical challenges: Chisel projects require Scala/SBT build systems, FIRRTL-to-Verilog compilation, and often have different verification methodologies compared to native Verilog projects. We are working to address these infrastructure requirements.
> > > >
> > > > We are committed to expanding HDL-FixBench and will release updates as new instances become available. We hope the reviewer may consider the foundational value of this contribution to the hardware AI community.

---

### Author Response · Authors · 2025-11-26
**General Comment for all Reviewers**

We thank the reviewers for their constructive feedback and for recognizing the importance of repository-level hardware bug repair with execution-grounded evaluation. We will revise the manuscript to incorporate the reviewers' suggestions and clarify points that may have been unclear. We are encouraged by the consensus on several core strengths:

- **First verifiable repository‑level benchmark for hardware.** Reviewers recognize that HDL‑FixBench "fills an important void in hardware LLM evaluation" by moving beyond component‑level tasks to repository‑scale challenges (8Fg3), addressing a "timely and underexplored problem" in automated HDL repair (Juiu).
- **Rigorous multi‑phase curation pipeline.** The combination of agent‑based semantic filtering and extensive Phase‑4 manual verification is described as demonstrating "a high degree of rigor" (suzP). Reviewers also view the base Docker image plus open‑sourced pipeline as a "realistic compromise" that maximizes reproducibility (8Fg3).
- **Diagnostic insights into hardware‑specific difficulty.** Reviewers appreciate that we do not "just report numbers" but identify concrete failure modes such as shallow hardware understanding and poor multi‑file coordination (8Fg3), providing "valuable insights on how HDL repair differs from traditional software engineering tasks" (Juiu).

Across the reviews, we identified several shared concerns, which we address below: Data Scarcity and Benchmark Scale, Patch-Size Thresholds and "Repository-Level" Scope.

### Data Scarcity and Discriminative Power

The 57-instance scale reflects fundamental constraints in the open-source HDL ecosystem:

- **Corpus-level scarcity.** In The Stack v2 (Lozhkov et al., 2024), a widely-used code corpus for LLM training, Verilog and SystemVerilog have 100–1000× fewer files than mainstream languages like Python or Java.
- **Repository-level scarcity.** Applying basic quality heuristics (>500 PRs, >100 stars) to GitHub yields fewer than ten viable HDL projects, compared to thousands for Python/Java/c++ under even stricter thresholds (>500 PRs, >3000 stars).
- **Rigorous per-instance curation.** Starting from ≈30K PRs across these repositories, our pipeline yields only 57 containerized FAIL→PASS tasks. Many promising PRs are unusable because base commits no longer build, bugs lack automated reproducers, or native regressions are too slow. Making each instance verifiable often requires days of expert effort—repairing dependencies, scripting DV flows, and sometimes writing new tests from scratch.

Despite the filtered scale, the benchmark provides initial evidence of discriminative power: models span 24-40% Resolved Rate (vs. 60-75% on SWE-bench Verified), and File-Level Precision varies by over 30 points. While we view these conclusions as preliminary given the limited scale, the results suggest that HDL-FixBench can meaningfully differentiate current agents and expose hardware-specific failure modes.

### Patch-Size Thresholds and Scope

**Rationale for patch-size thresholds.** During benchmark construction, we examined candidate PRs with larger patches and found that (i) nearly all correspond to refactorings or feature additions rather than bug fixes, and (ii) the few genuine large-patch bug fixes are uniformly unresolved by all agents. We plan to incorporate these as challenging instances.

**Scope and complexity of filtered tasks.**  We clarify that "repository-level" refers to the context and evaluation mode, agents interact with the full repository and run native simulation flows, not to patch size. The thresholds filter extreme refactorings while preserving the multi-file, cross-module nature of real RTL fixes. Even after filtering, HDL-FixBench retains a more complex distribution than SWE-bench Verified: 47% multi-file edits (vs. 14%) and 49% >20 lines (vs. 18%).

### Ongoing and planned expansion.

Since receiving the reviews, we have continued expanding HDL-FixBench along two axes:

- **Depth.** On existing repositories, we revisited failing PRs and found the benchmark already captures most accessible bug-fix cases; remaining candidates require days of expert effort.
- **Breadth.** We have begun applying our pipeline to additional repositories (other CORE-V cores, security-focused designs (Caliptra-RTL), Black-Parrot, bsg_manycore). Early experiments confirm the pipeline generalizes, but genuine RTL bug-fix PRs that yield verifiable FAIL→PASS tasks remain rare.

The current 57-instance benchmark therefore represents a carefully curated foundation that we are actively extending.

We thank the reviewers again for their time and constructive engagement. Reviewer-specific questions are addressed in our individual responses below.

---

### Author Response · Authors · 2025-12-04
**Summary of Revisions**

We thank all reviewers for their constructive feedback. This document summarizes the revisions made to the manuscript in response to the reviews.

---

## 1. Main Text Revisions

### Section 4.2: Per-Category Performance Analysis (Reviewer v2jx W4)

**Added Table 3**: A breakdown of resolved instances by bug category for each model.

This analysis reveals:
- Logic and Specification bugs remain far from solved (best models resolve ~40% and ~33%)
- Testbench/DV and Interface bugs serve as key differentiators across models
- Models exhibit distinct capability profiles (e.g., GPT-5 excels at TB/DV tasks while Kimi-K2 struggles with verification code)

### Limitations Section

**Scope of Repositories and Languages**: Clarified current scope (Verilog/SystemVerilog bug repair) and excluded areas (VHDL, Chisel, feature implementation, optimization). Added planned extensions: additional repositories and Chisel-based designs.

**Constraints in Task Curation and Scale**: Clarified that filtered instances still exhibit higher complexity than SWE-bench Verified. Explained Gemini 2.5 Pro selection rationale (most permissive for high recall). Added reference to cross-model validation in Appendix A.8.

**Reliance on Proprietary Toolchains**: Clarified that 14/57 tasks (Ibex + CVA6) can run on fully open-source tools. Added reference to Appendix A.7 for detailed breakdown.

---

## 2. New Appendix Sections

### A.5 Agent Scaffolding Configuration Details (Reviewer 8Fg3 W2, Reviewer Juiu Q1)

Provides comprehensive documentation of our hardware-specific agent configurations:
- **Prompt Information Content Per Iteration**: Details what information agents receive in initial and subsequent iterations
- **Available Tools and Action Space**: Tables 5-6 listing OpenHands and SWE-Agent tools for hardware debugging
- **Hardware-Specific System Prompts**: Complete prompt templates (Listings 2-3)
- **Runtime Environment Configuration**: Docker image contents and environment setup
- **Agent Interaction Loop**: Schematic of the iterative debugging process (Figure 6)

### A.6 Cost and Runtime Analysis (Reviewer Juiu Q2, Q3; Reviewer 8Fg3 Q7)

**Added Table 7**: API cost analysis per task instance
- Average input/output tokens and cost per case for all 7 models
- Key finding: GPT-5 demonstrates strong cost-effectiveness; DeepSeek-V3.1 offers lowest cost

**Added Table 8**: Runtime statistics per task instance
- Mean, median, and range of wall-clock times
- Reasoning mode indicator for each model
- Key finding: Reasoning-enabled models do not uniformly outperform non-reasoning ones

### A.7 EDA Toolchain Requirements (Reviewer 8Fg3 Q4, Q5, Q6)

**Added Table 9**: EDA toolchain requirements by repository
- Ibex (7 tasks) + CVA6 (7 tasks): Verilator only (fully open-source)
- OpenTitan (43 tasks): Synopsys VCS required
- Users without commercial licenses can evaluate on the 14-task open-source subset

### A.8 Cross-Model Validation of Phase-3 Semantic Filter (Reviewer suzP W2/Q2)

**Added Table 10**: Cross-model validation results

Addresses concerns about potential bias from using Gemini 2.5 Pro as the semantic filter:
- Phase-3 classification compared across Gemini-2.5-Pro, GPT-5, and Kimi-K2
- 72–85% three-way agreement across repositories
- Gemini is the most permissive (highest recall), ensuring we do not discard borderline RTL bugs

---

## 3. Minor Corrections

- Citation format: Verified consistent use of parenthetical citations throughout

---

## Summary of Changes by Reviewer Concern

| Reviewer | Concern | Response Location |
|----------|---------|-------------------|
| Juiu Q1 | Prompt content per iteration | Appendix A.5.1 |
| Juiu Q2 | API cost analysis | Appendix A.6.1, Table 7 |
| Juiu Q3 | Runtime and reasoning modes | Appendix A.6.2, Table 8 |
| Juiu Nit | Citation format | Verified throughout |
| suzP W2/Q2 | Filter bias validation | Appendix A.8, Table 10 |
| 8Fg3 W2 | Agent scaffolding details | Appendix A.5 |
| 8Fg3 Q4-Q6 | EDA tool requirements | Appendix A.7, Table 9 |
| 8Fg3 Q7 | Benchmark runtime | Appendix A.6.2, Table 8 |
| v2jx W4 | Bug type importance | Section 4.2, Table 3 |

---

We believe these revisions address the reviewers' concerns and strengthen the manuscript. We remain committed to further improving HDL-FixBench.

---

### Meta-Review · Area_Chair_MAYp · 2026-01-07

**Summary:**

This paper introduces HDL-FixBench, the first benchmark for evaluating LLMs on repository-level RTL bug repair. However, the submission raised significant concerns. Reviewers uniformly criticized the small scale and narrow domain focus, questioning statistical robustness and generalizability. Although the rebuttal clarified some aspects such as runtime tables, toolchain breakdowns, and per-category performance, it could not overcome the fundamental limitations of the proposed benchmark in terms of scale and diversity. Based on the above considerations, I recommend rejection, while encouraging the authors to further develop this promising direction.

**Reviewer Concerns:**

Addressed:
  - Clarification of agent scaffolding
  - Cross-model validation of semantic filter bias
  - Cost & runtime statistics
  - Per-category bug analysis

Still outstanding:
  - Benchmark scale & diversity
  - Patch-size threshold vs. “repository-level” claim
  - Non-LLM baselines
  - Contamination risk

**Reviewer Scores:**

- Juiu: keep positive
- suzP: 2 → 4
- 8Fg3: keep positive
- v2jx: 2 → 4

---

### Decision · Program_Chairs · 2026-01-26

Reject